METHODS AND RESOURCES

# Neural networks enable efficient and accurate simulation-based inference of evolutionary parameters from adaptation dynamics

Grace Avecilla[1,2], Julie N. Chuong[1,2], Fangfei Li[3], Gavin Sherlock[3], David Gresham[1,2]*, Yoav Ram[4]*

**1** Department of Biology, New York University, New York, New York, United States of America, **2** Center for Genomics and Systems Biology, New York University, New York, New York, United States of America, **3** Department of Genetics, Stanford University, California, Stanford, United States of America, **4** School of Zoology, Faculty of Life Sciences, Tel Aviv University, Tel Aviv, Israel

* dgresham@nyu.edu (DG); yoav@yoavram.com (YR)

**Data Availability Statement:** All source code for performing the analyses and reproducing the figures is available at https://github.com/graceave/

## Abstract

The rate of adaptive evolution depends on the rate at which beneficial mutations are introduced into a population and the fitness effects of those mutations. The rate of beneficial mutations and their expected fitness effects is often difficult to empirically quantify. As these 2 parameters determine the pace of evolutionary change in a population, the dynamics of adaptive evolution may enable inference of their values. Copy number variants (CNVs) are a pervasive source of heritable variation that can facilitate rapid adaptive evolution. Previously, we developed a locus-specific fluorescent CNV reporter to quantify CNV dynamics in evolving populations maintained in nutrient-limiting conditions using chemostats. Here, we use CNV adaptation dynamics to estimate the rate at which beneficial CNVs are introduced through de novo mutation and their fitness effects using simulation-based likelihood–free inference approaches. We tested the suitability of 2 evolutionary models: a standard Wright–Fisher model and a chemostat model. We evaluated 2 likelihood-free inference algorithms: the well-established Approximate Bayesian Computation with Sequential Monte Carlo (ABC-SMC) algorithm, and the recently developed Neural Posterior Estimation (NPE) algorithm, which applies an artificial neural network to directly estimate the posterior distribution. By systematically evaluating the suitability of different inference methods and models, we show that NPE has several advantages over ABC-SMC and that a Wright–Fisher evolutionary model suffices in most cases. Using our validated inference framework, we estimate the CNV formation rate at the *GAP1* locus in the yeast *Saccharomyces cerevisiae* to be $10^{-4.7}$ to $10^{-4}$ CNVs per cell division and a fitness coefficient of 0.04 to 0.1 per generation for *GAP1* CNVs in glutamine-limited chemostats. We experimentally validated our inference-based estimates using 2 distinct experimental methods—barcode lineage tracking and pairwise fitness assays—which provide independent confirmation of the accuracy of our approach. Our results are consistent with a beneficial CNV supply rate that is 10-fold greater than the estimated rates of beneficial single-nucleotide mutations, explaining the outsized importance of CNVs in rapid adaptive evolution. More generally, our study demonstrates the utility of novel neural network–based likelihood–free inference methods for inferring the

cnv_sims_inference. All of the data can be found at https://osf.io/e9d5x/.

**Funding:** This work was supported in part by grants from the Israel Science Foundation (552/19) and Minerva Stiftung Center for Lab Evolution (YR), from the NIH (R01 GM134066 and R01 GM107466) (DG) and NSF (MCB1818234) (DG), from the NIH (R35 GM131824 and R01 AI136992) (GS), NSF GRFP (DGE1342536) (GA) and (DGE1839302) (JC), and NIH (T32 GM132037) (JC). The funders had no role in study design, data collection and analysis, decision to publish, or preparation of the manuscript.

**Competing interests:** The authors have declared that no competing interests exist.

**Abbreviations:** ABC, Approximate Bayesian Computation; ABC-SMC, Approximate Bayesian Computation with Sequential Monte Carlo; AIC, Akaike information criterion; CNV, copy number variant; DFE, distribution of fitness effects; HDI, highest density interval; HDR, highest density region; KDE, kernel density estimate; MAF, masked autoregressive flow; MAP, maximum a posteriori; NPE, Neural Posterior Estimation; NSF, neural spline flow; RMSE, root mean square error; WAIC, widely applicable information criterion.

rates and effects of evolutionary processes from empirical data with possible applications ranging from tumor to viral evolution.

## Introduction

Evolutionary dynamics are determined by the supply rate of beneficial mutations and their associated fitness effect. As the combination of these 2 parameters determines the overall rate of adaptive evolution, experimental methods are required for separately estimating them. The fitness effects of beneficial mutations can be determined using competition assays [1,2], and mutation rates are typically estimated using mutation accumulation or Luria–Delbrück fluctuation assays [1,3]. An alternative approach to estimating both the rate and effect of beneficial mutations entails quantifying the dynamics of adaptive evolution and using statistical inference methods to find parameter values that are consistent with the dynamics [4–7]. Approaches to measure the dynamics of adaptive evolution, quantified as changes in the frequencies of beneficial alleles, have become increasingly accessible using either phenotypic markers [8] or high-throughput DNA sequencing [9]. Thus, inference methods using adaptation dynamics data hold great promise for determining the underlying evolutionary parameters.

Fitness effects of beneficial mutations comprise a portion of a distribution of fitness effects (DFE). Determining the parameters of the DFE in a given condition is a central goal of evolutionary biology. Typically, beneficial mutations can occur at multiple loci and thus variance in the DFE reflects genetic heterogeneity. However, in some scenarios, a single locus is the dominant gene in which beneficial mutations occur, such as the case of mutations in the $\beta$-lactamase gene underlying $\beta$-lactam antibiotic resistance or in *rpoB* underlying rifampicin resistance in bacteria [10,11]. In this case, different mutations at the same locus confer differential beneficial effects resulting in a locus-specific DFE. Typically, a DFE of beneficial mutations encompasses both allelic and locus heterogeneity.

Copy number variants (CNVs) are defined as deletions or amplifications of genomic sequences. Due to their high rate of formation and strong fitness effects, they can underlie rapid adaptive evolution in diverse scenarios ranging from niche adaptation to speciation [12–16]. In the short term, CNVs may provide immediate fitness benefits by altering gene dosage. Over longer evolutionary timescales, CNVs can provide the raw material for the generation of evolutionary novelty through diversification of different gene copies [17]. As a result, CNVs are common in human populations [18–20], domesticated and wild populations of animals and plants [21–23], pathogenic and nonpathogenic microbes [24–27], and viruses [28–30]. CNVs can be both a driver and a consequence of cancers (reviewed in [31]).

Although critically important to adaptive evolution, our understanding of the dynamics and reproducibility of CNVs in adaptive evolution is poor. Specifically, key evolutionary properties of CNVs, including their rate of formation and fitness effects, are largely unknown. As with other classes of genomic variation, CNV formation is a relatively rare event, occurring at sufficiently low frequencies to make experimental measurement challenging. Estimates of de novo CNV rates are derived from indirect and imprecise methods, and even when genome-wide mutation rates are directly quantified by mutation accumulation studies and whole-genome sequencing, estimates depend on both genotype and condition [3] and vary by orders of magnitude [32–39].

Fitness effects of CNVs vary depending on gene content, genetic background, and the environment. In evolution experiments in many systems, CNVs arise repeatedly in response to

strong selection [40–47], consistent with strong beneficial fitness effects. Several of these studies measured fitness of clonal isolates containing CNVs and reported selection coefficients ranging from −0.11 to 0.6 [40,47,48]. However, the fitness of lineages containing CNVs varies between isolates even within studies, which could be due to additional heritable variation or to differences in fitness between different types of CNVs (e.g., aneuploidy versus single-gene amplification).

Due to the challenge of empirically measuring rates and effects of beneficial mutations across many genetic backgrounds, conditions, and types of mutations, researchers have attempted to infer these parameters from population-level data using evolutionary models and Bayesian inference [5,6,49]. This approach has several advantages. First, model-based inference provides estimations of interpretable parameters and the opportunity to compare multiple models. Second, the degree of uncertainty associated with a point estimate can be quantified. Third, a posterior distribution over model parameters allows exploration of parameter combinations that are consistent with the observed data, and posterior distributions can provide insight into certain relationships between parameters [50]. Fourth, posterior predictions can be generated using the model and either compared to the data or used to predict the outcome of differing scenarios.

Standard Bayesian inference requires a likelihood function, which gives the probability of obtaining the observed data given some values of the model parameters. However, for many evolutionary models, such as the Wright–Fisher model, the likelihood function is analytically and/or computationally intractable. Likelihood-free simulation-based Bayesian inference methods that bypass the likelihood function, such as Approximate Bayesian Computation (ABC; [51]), have been developed and used extensively in population genetics [52,53], ecology and epidemiology [54,55], cosmology [56], as well as experimental evolution [4,6,57–59]. The simplest form of likelihood-free inference is rejection ABC [60,61], in which model parameter proposals are sampled from a prior distribution, simulations are generated based on those parameter proposals, and simulated data are compared to empirical observations using summary statistics and a distance function. Proposals that generate simulated data with a distance less than a defined tolerance threshold are considered samples from the posterior distribution and can therefore be used for its estimation. Efficient sampling methods have been introduced, namely Markov chain Monte Carlo [62] and Sequential Monte Carlo (SMC) [63], which iteratively select proposals based on previous parameters samples so that regions of the parameter space with higher posterior density are explored more often. A shortcoming of ABC is that it requires summary statistics and a distance function, which may be difficult to choose appropriately and compute efficiently, especially when using high-dimensional or multimodal data, although methods have been developed to address this challenge [52,64,65].

Recently, new inference methods have been introduced that directly approximate the likelihood or the posterior density function using deep neural density estimators—artificial neural networks that approximate density functions. These methods, which have recently been used in neuroscience [50], population genetics [66], and cosmology [67], forego the summary and distance functions, can use data with higher dimensionality, and perform inference more efficiently [50,67,68].

Despite being originally developed to analyze population genetic data, e.g., to infer parameters of the coalescent model [60–63], likelihood-free methods have only been used in a small number of experimental evolution studies. Hegreness and colleagues [5] estimated the rate and mean fitness effect of beneficial mutations in *Escherichia coli*. They performed 72 replicates of a serial dilution evolution experiment, starting with equal frequencies of 2 strains that differ only in a fluorescent marker in a putatively neutral location and allowed them to evolve over 300 generations. Following the marker frequencies, they estimated from each

experimental replicate 2 summary statistics: the time when a beneficial mutation starts to spread in the population and the rate at which its frequency increases. They then ran 500 simulations of an evolutionary model using a grid of model parameters to produce a theoretical distribution of summary statistics. Finally, they used the one-dimensional Kolmogorov–Smirnov distance between the empirical and theoretical summary statistic distributions to assess the inferred parameters. Barrick and colleagues [6] also inferred the rate and mean fitness effect from similar serial dilution experiments using a different evolutionary model implemented with a τ-leap stochastic simulation algorithm. They used the same summary statistics but applied the two-dimensional Kolmogorov–Smirnov distance function to better account for dependence between the summary statistics. de Sousa and colleagues [69] also focused on evolutionary experiments with 2 neutral markers. Their model included 3 parameters: the beneficial mutation rate and the 2 parameters of a Gamma distribution for the fitness effects of beneficial mutations. They introduced a new summary statistic that uses both the marker frequency trajectories and the population mean fitness trajectories (measured using competition assays). They summarized these data by creating histograms of the frequency values and fitness values for each of 6 time points. This resulted in 66 summary statistics necessitating the application of a regression-based method to reduce the dimensionality of the summary statistics and achieve greater efficiency [65,69]. A simpler approach was taken by Harari and colleagues [49], who used a rejection ABC approach to estimate a single model parameter, the endoreduplication rate, from evolutionary experiments with yeast. They used the frequency dynamics of 3 genotypes (haploid and diploid homozygous and heterozygous at the *MAT* locus) without a summary statistic. The distance between the empirical results and 100 simulations was computed as the mean absolute error. Recently, Schenk and colleagues [69] inferred the mean mutation rate and fitness effect for 3 classes of mutations from serial dilution experiments at 2 different population sizes, which they sequenced at the end of the experiment. They used a Wright–Fisher model to simulate the frequency of fixed mutations in each class and used a neural network approach to estimate the parameters that best fit their data. These prior studies point to the potential of simulation-based inference.

Previously, we developed a fluorescent CNV reporter system in the budding yeast, *Saccharomyces cerevisiae*, to quantify the dynamics of de novo CNVs during adaptive evolution [48]. Using this system, we quantified CNV dynamics at the *GAP1* locus, which encodes a general amino acid permease, in nitrogen-limited chemostats for over 250 generations in multiple populations. We found that *GAP1* CNVs reproducibly arise early and sweep through the population. By combining the *GAP1* CNV reporter with barcode lineage tracking and whole-genome sequencing, we found that $10^2$ to $10^4$ independent CNV-containing lineages comprising diverse structures compete within populations.

In this study, we estimate the formation rate and fitness effect of *GAP1* CNVs. We tested both ABC-SMC [70] and a neural density estimation method, Neural Posterior Estimation (NPE) [71], using a classical Wright–Fisher model [72] and a chemostat model [73]. Using simulated data, we tested the utility of the different evolutionary models and inference methods. We find that NPE has better performance than ABC-SMC. Although a more complex model has improved performance, the simpler and more computationally efficient Wright–Fisher model is appropriate in most scenarios. We validated our approach by comparison to 2 different experimental methods: lineage tracking and pairwise fitness assays. We estimate that in glutamine-limited chemostats, beneficial *GAP1* CNVs are introduced at a rate of $10^{-4.7}$ to $10^{-4}$ per cell division and have a selection coefficient of 0.04 to 0.1 per generation. NPE is likely to be a useful method for inferring evolutionary parameters across a variety of scenarios, including tumor and viral evolution, providing a powerful approach for combining experimental and computational methods.

## Results

In a previous experimental evolution study, we quantified the dynamics of de novo CNVs in 9 populations using a prototrophic yeast strain containing a fluorescent *GAP1* CNV reporter. [48]. Populations were maintained in glutamine-limited chemostats for over 250 generations and sampled every 8 to 20 generations (25 time points in total) to determine the proportion of cells containing a *GAP1* CNV using flow cytometry (populations gln_01-gln_09 in **Fig 1A**). In the same study, we also performed 2 replicate evolution experiments using the fluorescent *GAP1* CNV reporter and lineage-tracking barcodes quantifying the proportion of the population with a *GAP1* CNV at 32 time points (populations bc01-bc02 in **Fig 1A**) [48]. We used interpolation to match time points between these 2 experiments (**S1 Fig**) resulting in a dataset comprising the proportion of the population with a *GAP1* CNV at 25 time points in 11

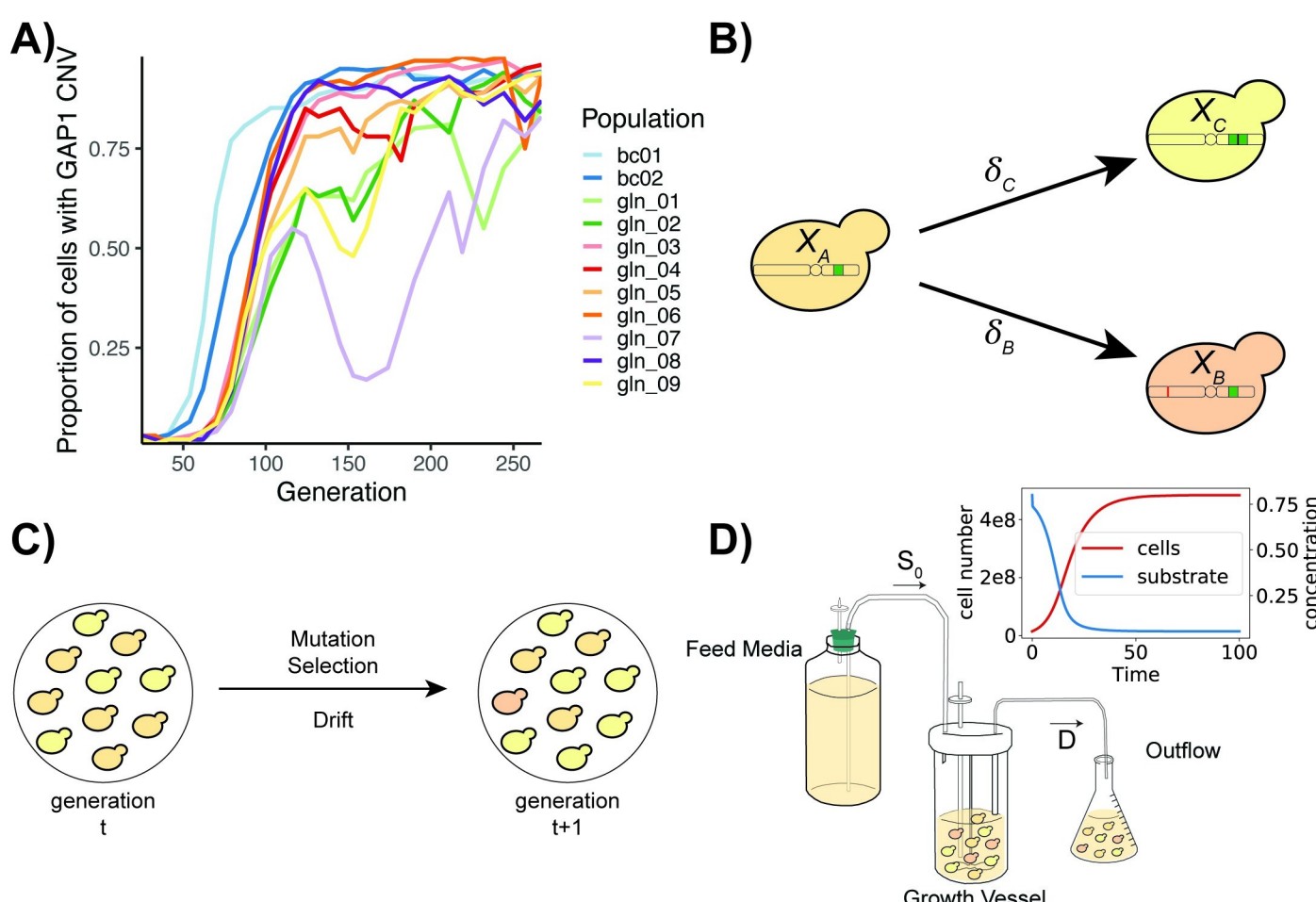

**Fig 1. Empirical data and evolutionary models. (A)** Estimates of the proportion of cells with *GAP1* CNVs for 11 *S. cerevisiae* populations containing either a fluorescent *GAP1* CNV reporter (gln_01 to gln_09) or a fluorescent *GAP1* CNV reporter and lineage tracking barcodes (bc01 and bc02) evolving in glutamine-limited chemostats, from [48]. **(B)** In our models, cells with the ancestral genotype ($X_A$) can give rise to cells with a *GAP1* CNV ($X_C$) or other beneficial mutation ($X_B$) at rates $\delta_C$ and $\delta_B$, respectively. **(C)** The WF model has discrete, nonoverlapping generations and a constant population size. Allele frequencies in the next generation change from the previous generation due to mutation, selection, and drift. **(D)** In the chemostat model, medium containing a defined concentration of a growth-limiting nutrient ($S_0$) is added to the culture at a constant rate. The culture, containing cells and medium, is removed by continuous dilution at rate *D*. Upon inoculation, the number of cells in the growth vessel increases and the limiting-nutrient concentration decreases until a steady state is reached (red and blue curves in inset). Within the growth vessel, cells grow in continuous, overlapping generations undergoing mutation, selection, and drift. Data and code required to generate **A** can be found at https://doi.org/10.17605/OSF.IO/E9D5X. CNV, copy number variant; WF, Wright–Fisher.

replicate evolution experiments. In this study, we tested whether the observed dynamics of CNV-mediated evolution provide a means of inferring the underlying evolutionary parameters.

## Overview of evolutionary models

We tested 2 models of evolution: the classical Wright–Fisher model [72] and a specialized chemostat model [73]. Previously, it has been shown that a single effective selection coefficient may be sufficient to model evolutionary dynamics in populations undergoing adaptation [5]. Therefore, we focus on beneficial mutations and assume a single selection coefficient for each class of mutation. In both models, we start with an isogenic population in which *GAP1* CNV mutations occur at a rate $\delta_C$ and other beneficial mutations occur at rate $\delta_B$ (**Fig 1B**). In our simulations, cells can acquire only a single beneficial mutation, either a CNV at *GAP1* or some other beneficial mutation (i.e., single nucleotide variant, transposition, diploidization, or CNV at another locus). In all simulations (except for sensitivity analysis, see the "Inference from empirical evolutionary dynamics" section), the formation rate of beneficial mutations other than *GAP1* CNVs was fixed at $\delta_B = 10^{-5}$ per genome per cell division, and the selection coefficient was fixed at $s_B = 0.001$, based on estimates from previous experiments using yeast in several conditions [74–76]. Our goal was to infer the *GAP1* CNV formation rate, $\delta_C$, and *GAP1* CNV selection coefficient, $s_C$.

The 2 evolutionary models have several unique features. In the Wright–Fisher model, the population size is constant, and each generation is discrete. Therefore, genetic drift is efficiently modeled using multinomial sampling (**Fig 1C**). In the chemostat model [73], fresh medium is added to the growth vessel at a constant rate and medium, and cells are removed from the growth vessel at the same rate resulting in continuous dilution of the culture (**Fig 1D**). Individuals are randomly removed from the population through the dilution process, regardless of fitness, in a manner analogous to genetic drift. In the chemostat model, we start with a small initial population size and a high initial concentration of the growth-limiting nutrient. Following inoculation, the population size increases and the growth-limiting nutrient concentration decreases until a steady state is attained that persists throughout the experiment. As generations are continuous and overlapping in the chemostat model, we use the Gillespie algorithm with τ-leaping [77] to simulate the population dynamics. Growth parameters in the chemostat are based on experimental conditions during the evolution experiments [48] or taken from the literature (**Table 1**).

**Table 1. Chemostat parameters.**

| Parameter | Value | Source |
|---|---|---|
| $k_A = k_B = k_C$ | 0.103 mM | Airoldi and colleagues (2016) https://doi.org/10.1091/mbc.E14-05-1013 |
| $Y_A = Y_B = Y_C$ | 32,445,000 cells/mL/mM nitrogen | Airoldi and colleagues (2016) https://doi.org/10.1091/mbc.E14-05-1013 |
| Expected *S* at steady state | Approximately 0.08 mM | Airoldi and colleagues (2016) https://doi.org/10.1091/mbc.E14-05-1013 |
| $\mu_{max}$ | 0.35 hour$^{-1}$ | Cooper TG (1982) Nitrogen metabolism in *S. cerevisiae* |
| $D$ | 0.12 hour$^{-1}$ | Lauer and colleagues (2018) https://doi.org/10.1371/journal.pbio.3000069 |
| $S_0$ | 0.8 mM | Lauer and colleagues (2018) https://doi.org/10.1371/journal.pbio.3000069 |
| Expected cell density at steady state | Approximately $2.5 \times 10^7$ cells/mL | Lauer and colleagues (2018) https://doi.org/10.1371/journal.pbio.3000069 |
| Doubling time | 5.8 hours | Lauer and colleagues (2018) https://doi.org/10.1371/journal.pbio.3000069 |

## Overview of inference strategies

We tested 2 likelihood-free Bayesian methods for joint inference of the *GAP1* CNV formation rate and the *GAP1* CNV fitness effect: Approximate Bayesian Computation with Sequential Monte Carlo (ABC-SMC) [63] and NPE [78–80]. We used the proportion of the population with a *GAP1* CNV at 25 time points as the observed data (**Fig 1A**). For both methods, we defined a log-uniform prior distribution for the CNV formation rate ranging from $10^{-12}$ to $10^{-3}$ and a log-uniform prior distribution for the selection coefficient ranging from $10^{-4}$ to 0.4.

We applied ABC-SMC (**Fig 2A**), implemented in the Python package *pyABC* [70]. We used an adaptively weighted Euclidean distance function to compare simulated data to observed data. Thus, the distance function adapts over the course of the inference process based on the amount of variance at each time point [81]. The number of samples drawn from the proposal distribution (and therefore number of simulations) is changed at each iteration of the ABC-SMC algorithm using the adaptive population strategy, which is based on the shape of the current posterior distribution [82]. We applied bounds on the maximum number of samples used to approximate the posterior in each iteration; however, the total number of samples (simulations) used in each iteration is greater because not all simulations are accepted for posterior estimation (see **Methods**). For each observation, we performed ABC-SMC with multiple iterations until either the acceptance threshold ($\varepsilon = 0.002$) was reached or until 10 iterations had been completed. We performed inference on each observation independently 3 times. Although we refer to different observations belonging to the same "training set," a different ABC-SMC procedure must be performed for each observation.

We applied NPE (**Fig 2B**), implemented in the Python package *sbi* [71], and tested 2 specialized normalizing flows as density estimators: a masked autoregressive flow (MAF) [83] and a neural spline flow (NSF) [84]. The normalizing flow is used as a density estimator to "learn" an amortized posterior distribution, which can then be evaluated for specific observations. Thus, amortization allows for evaluation of the posterior for each new observation without the need to retrain the neural network. To test the sensitivity of our inference results on the set of simulations used to learn the amortized posterior, we trained 3 independent amortized networks with different sets of simulations generated from the prior distribution and compared our resulting posterior distributions for each observation. We refer to inferences made with the same amortized network as having the same "training set."

## NPE outperforms ABC-SMC

To test the performance of each inference method and evolutionary model, we generated 20 simulated synthetic observations for each model (Wright–Fisher or chemostat) over 4 combinations of CNV formation rates and selection coefficients, resulting in 40 synthetic observations (i.e., 5 simulated observations per combination of model, $\delta_C$, and $s_C$). We refer to the parameters that generated the synthetic observation as the "true" parameters. For each synthetic observation, we performed inference using each method 3 times. Inference was performed using the same evolutionary model as that used to generate the observation. We found that NPE using NSF as the density estimator was superior to NPE using MAF, and, therefore, we report results using NSF in the main text (results using MAF are in **S2 Fig**).

For each inference method, we plotted the joint posterior distribution with the 50% and 95% highest density regions (HDR) [85] demarcated (**Fig 2C**, **S1 Data** in https://doi.org/10. 17605/OSF.IO/E9D5X). The true parameters are expected to be covered by these HDRs at least 50% and 95% of the time, respectively. We also computed the marginal 95% highest density intervals (HDIs) [85] using the marginal posterior distributions for the *GAP1* CNV selection coefficient and *GAP1* CNV formation rate. We found that the true parameters were

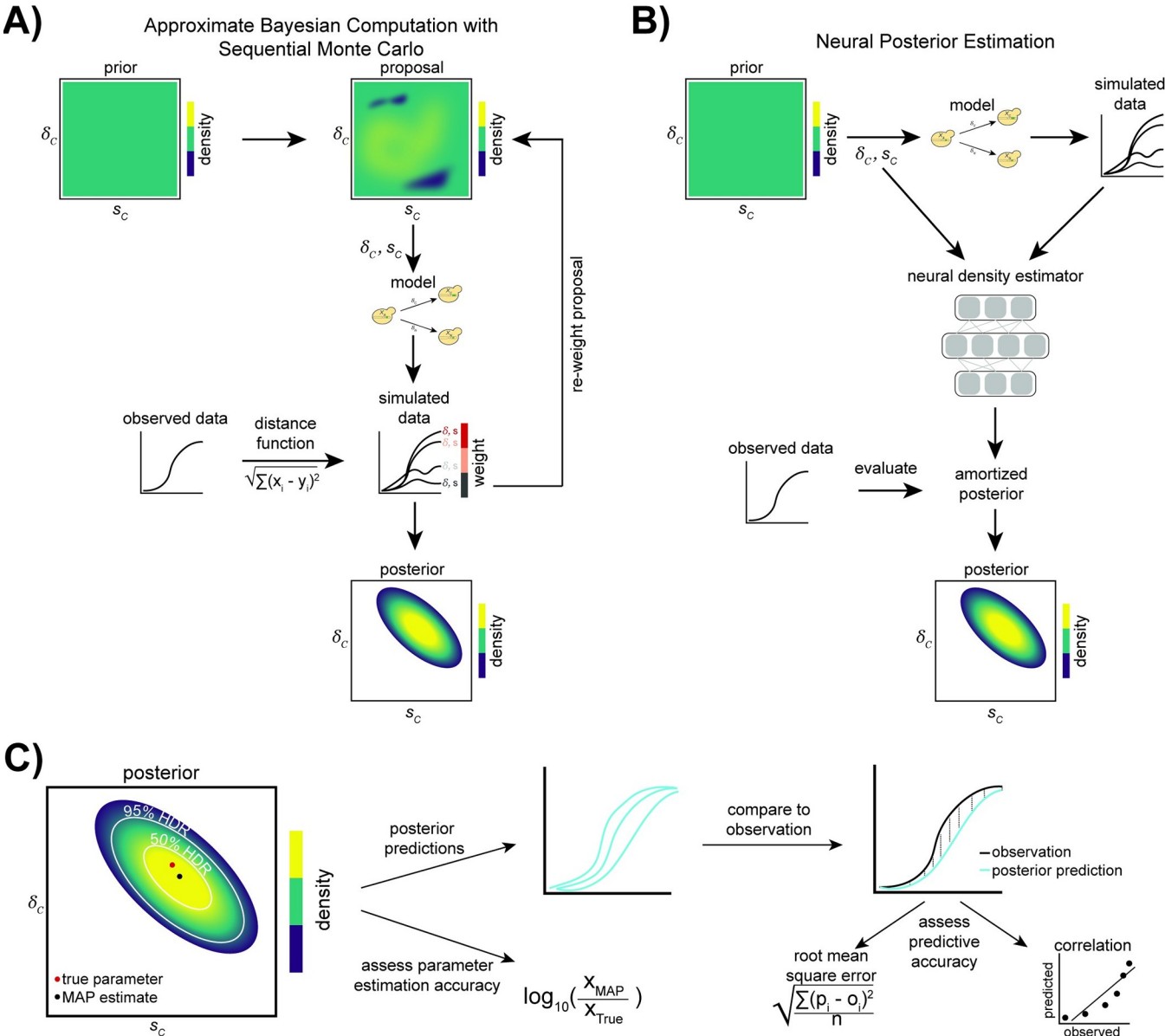

**Fig 2. Inference methods and performance assessment.** (**A**) When using ABC-SMC, in the first iteration, a proposal for the parameters $\delta_C$ (*GAP1* CNV formation rate) and $s_C$ (*GAP1* CNV selection coefficient) is sampled from the prior distribution. Simulated data are generated using either a WF or chemostat model and the current parameter proposal. The distance between the simulated data and the observed data is computed, and the proposed parameters are weighted by this distance. These weighted parameters are used to sample the proposed parameters in the next iteration. Over many iterations, the weighted parameter proposals provide an increasingly better approximation of the posterior distribution of $\delta_C$ and $s_C$ (adapted from [68]). (**B**) In NPE, simulated data are generated using parameters sampled from the prior distribution. From the simulated data and parameters, a density-estimating neural network learns the joint density of the model parameters and simulated data (the "amortized posterior"). The network then evaluates the conditional density of model parameters given the observed data, thus providing an approximation of the posterior distribution of $\delta_C$ and $s_C$ (adapted from [50,68].) (**C**) Assessment of inference performance. The 50% and 95% HDRs are shown on the joint posterior distribution with the true parameters and the MAP parameter estimates. We compare the true parameters to the estimates by their log ratio. We also generate posterior predictions (sampling 50 parameters from the joint posterior distribution and using them to simulate frequency trajectories, $\rho_i$), which we compare to the observation, $o_i$, using the RMSE and the correlation coefficient. ABC-SMC, Approximate Bayesian Computation with Sequential Monte Carlo; CNV, copy number variant; HDR, highest density region; MAP, maximum a posteriori; NPE, Neural Posterior Estimation; RMSE, root mean square error; WF, Wright–Fisher.

within the 50% HDR in half or more of the tests (averaged over 3 training sets) across a range of parameter values with the exception of ABC-SMC applied to the Wright–Fisher model when the *GAP1* CNV formation rate ($\delta_C = 10^{-7}$) and selection coefficient ($s_C = 0.001$) were

both low (**Fig 3A**). The true parameters were within the 95% HDR in 100% of tests (**S1 Data** in https://doi.org/10.17605/OSF.IO/E9D5X). The width of the HDI is informative about the degree of uncertainty associated with the parameter estimation. The HDIs for both fitness effect and formation rate tend to be smaller when inferring with NPE compared to ABC-SMC, and this advantage of NPE is more pronounced when the CNV formation rate is high ($\delta_C = 10^{-5}$) (**Fig 3B and 3C**).

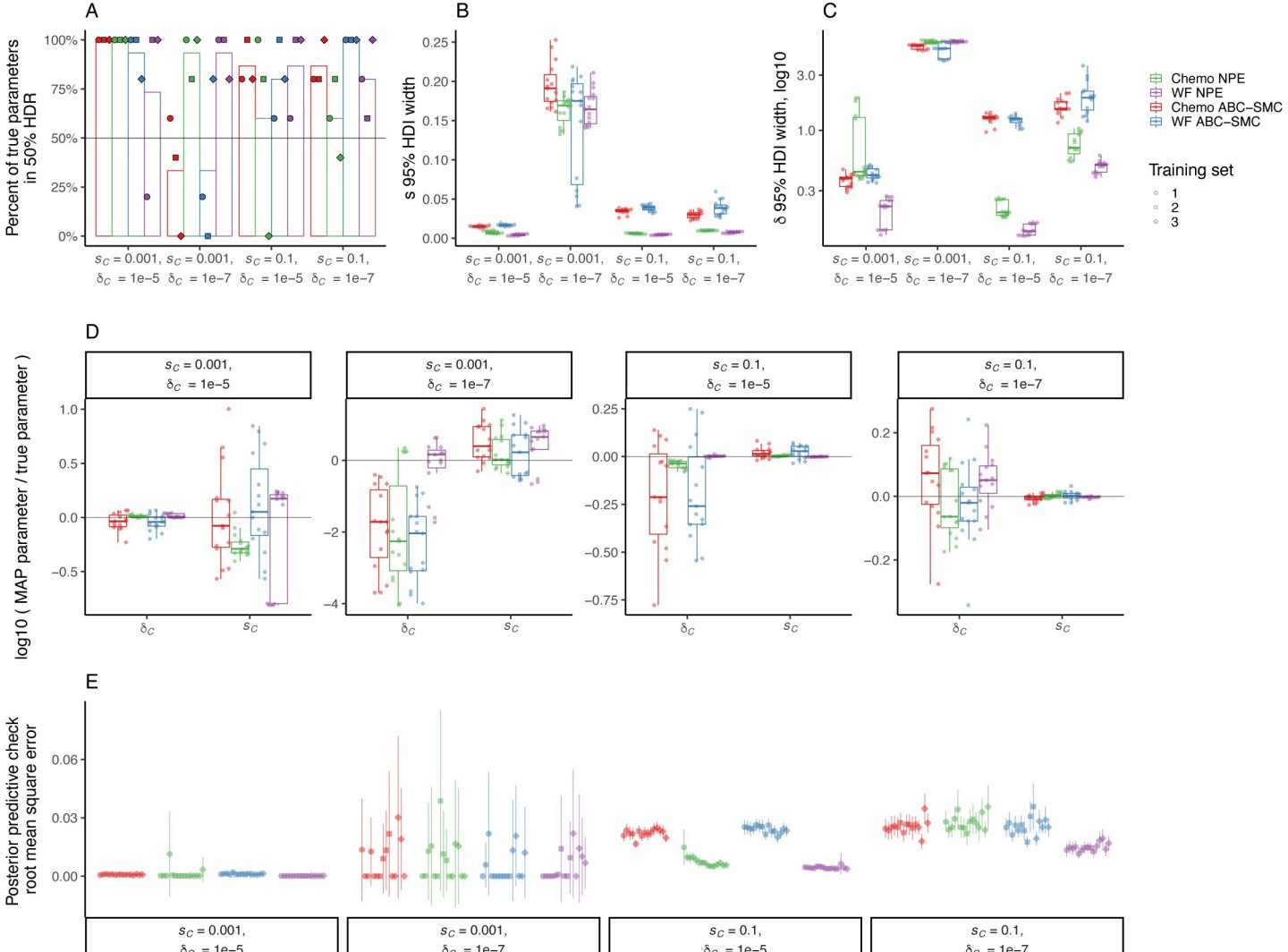

**Fig 3. Performance assessment of inference methods using simulated synthetic observations.** The figure shows the results of inference on 5 simulated synthetic observations using either the WF or chemostat (Chemo) model per combination of fitness effect $s_C$ and formation rate $\delta_C$. Simulations and inference were performed using the same model. For NPE, each training set corresponds to an independently amortized posterior distribution trained on a different set of 100,000 simulations, with which each synthetic observation was evaluated to produce a separate posterior distribution. For ABC-SMC, each training set corresponds to independent inference procedures on each observation with a maximum of 10,000 total simulations accepted for each inference procedure and a stopping criteria of 10 iterations or $\varepsilon < = 0.002$, whichever occurs first. **(A)** The percent of true parameters covered by the 50% HDR of the inferred posterior distribution. The bar height shows the average of 3 training sets. Horizontal line marks 50%. **(B, C)** Distribution of widths of 95% HDI of the posterior distribution of the fitness effect $s_C$ (B) and CNV formation rate $\delta_C$ (C), calculated as the difference between the 97.5 percentile and 2.5 percentile, for each separately inferred posterior distribution. **(D)** Log ratio of MAP estimate to true parameter for $s_C$ and $\delta_C$. Note the different y-axis ranges. Gray horizontal line represents a log ratio of zero, indicating an accurate MAP estimate. **(E)** Mean and 95% confidence interval of RMSE of 50 posterior predictions compared to the synthetic observation from which the posterior was inferred. Data and code required to generate this figure can be found at https://doi.org/10.17605/OSF.IO/E9D5X. ABC-SMC, Approximate Bayesian Computation with Sequential Monte Carlo; CNV, copy number variant; HDI, highest density interval; HDR, highest density region; MAP, maximum a posteriori; NPE, Neural Posterior Estimation; RMSE, root mean square error; WF, Wright–Fisher.

We computed the maximum a posteriori (MAP) estimate of the *GAP1* CNV formation rate and selection coefficient by determining the mode (i.e., argmax) of the joint posterior distribution, and computed the log ratio of the MAP relative to the true parameters. We find that the MAP estimate is close to the true parameter (i.e., the log ratio is close to zero) when the selection coefficient is high ($s_C = 0.1$), regardless of the model or method, and much of the error is due to the formation rate estimation error (**Fig 3D**). Generally, the MAP estimate is within an order of magnitude of the true parameter (i.e., the log ratio is less than 1), except when the formation rate and selection coefficient are both low ($\delta_C = 10^{-7}$, $s_C = 0.001$); in this case, the formation rate was underestimated up to 4-fold, and the selection coefficient was slightly overestimated (**Fig 3D**). In some cases, there are substantial differences in log ratio between training sets using NPE; however, this variation in log ratio is usually less than the variation in the log ratio when performing inference with ABC-SMC. Overall, the log ratio tends to be closer to zero (i.e., estimate close to true parameter) when using NPE (**Fig 3D**).

We performed posterior predictive checks by simulating *GAP1* CNV dynamics using the MAP estimates as well as 50 parameter values sampled from the posterior distribution (**S1 Data** in https://doi.org/10.17605/OSF.IO/E9D5X). We computed both the root mean square error (RMSE) and the correlation coefficient between posterior predictions and the observation to measure the prediction accuracy (**Fig 3E, S3 Fig**). We find that the RMSE posterior predictive accuracy of NPE is similar to, or better than, that of ABC-SMC (**Fig 3E**). The predictive accuracy quantified using correlation was close to 1 for all cases except when *GAP1* CNV formation rate and selection coefficient are both low ($s_C = 0.001$ and $\delta_C = 10^{-7}$) (**S3 Fig**).

We performed model comparison using both Akaike information criterion (AIC), computed using the MAP estimate, and widely applicable information criterion (WAIC), computed over the entire posterior distribution [86]. Lower values imply higher predictive accuracy and a difference of 2 is considered significant (**S4 Fig**) [87]. We find similar results for both criteria: NPE with either model have similar values, although the value for Wright–Fisher is sometimes slightly lower than the value for the chemostat model. When $s_C = 0.1$, the value for NPE is consistently and significantly lower than for ABC-SMC. When $\delta_C = 10^{-5}$ and $s_C = 0.001$, the value for NPE with the Wright–Fisher model is significantly lower than that for ABC-SMC, while the NPE with the chemostat model is not. The difference between any combination of model and method was insignificant for $\delta_C = 10^{-7}$ and $s_C = 0.001$. Therefore, NPE is similar or better than ABC-SMC using either evolutionary model and for all tested combinations of *GAP1* CNV formation rate and selection coefficient, and we further confirmed the generality of this trend using the Wright–Fisher model and 8 additional parameter combinations (**S5 Fig**).

We performed NPE using 10,000 or 100,000 simulations to train the neural network and found that increasing the number of simulations did not substantially reduce the MAP estimation error, but did tend to decrease the width of the 95% HDIs for both parameters (**S6 Fig**). Similarly, we performed ABC-SMC with per observation maximum accepted parameter samples (i.e., "particles" or "population size") numbers of 10,000 and 100,000, which correspond to increasing number of simulations per inference procedure, and found that increasing the budget decreases the widths of the 95% HDIs for both parameters (**S6 Fig**). Overall, amortization with NPE allowed for more accurate inference using fewer simulations corresponding to less computation time (**S7 Fig**).

## The Wright–Fisher model is suitable for inference using chemostat dynamics

Whereas the chemostat model is a more precise description of our evolution experiments, both the model itself and its computational implementation have some drawbacks. First, the

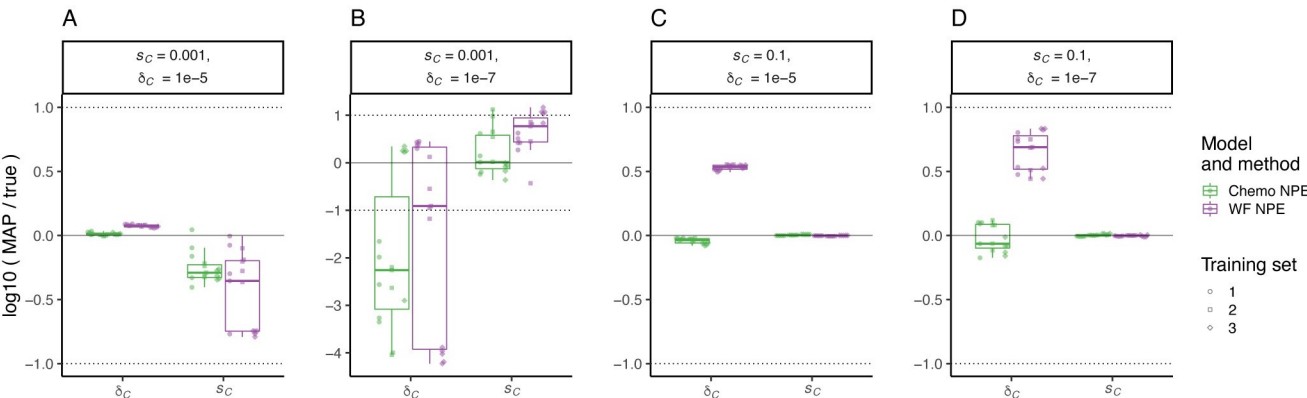

**Fig 4. Inference with WF model from chemostat dynamics.** The figure shows results of inference using NPE and either the WF or chemostat (Chemo) model on 5 simulated synthetic observations generated using the chemostat model for different combinations of fitness effect $s_C$ and formation rate $\delta_C$. Boxplots and markers show the log ratio of MAP estimate to true parameters for $s_C$ and $\delta_C$. Horizontal solid line represents a log ratio of zero, indicating an accurate MAP estimate; dotted lines indicate an order of magnitude difference between the MAP estimate and the true parameter. Data and code required to generate this figure can be found at https://doi.org/10.17605/OSF.IO/E9D5X. MAP, maximum a posteriori; NPE, Neural Posterior Estimation; WF, Wright–Fisher.

model is a stochastic continuous time model implemented using the τ-leap method [77]. In this method, time is incremented in discrete steps and the number of stochastic events that occur within that time step is sampled based on the rate of events and the system state at the previous time step. For accurate stochastic simulation, event rates and probabilities must be computed at each time step, and time steps must be sufficiently small. This incurs a heavy computational cost as time steps are considerably smaller than one generation, which is the time step used in the simpler Wright–Fisher model. Moreover, the chemostat model itself has additional parameters compared to the Wright–Fisher model, which must be experimentally measured or estimated.

The Wright–Fisher model is more general and more computationally efficient than the chemostat model (**S1 Table**). Therefore, we investigated if it can be used to perform accurate inference with NPE on synthetic observations generated by the chemostat model. By assessing how often the true parameters were covered by the HDRs, we found that the Wright–Fisher is a good enough approximation of the full chemostat dynamics when selection is weak ($s_C =$ 0.001) (**S8 Fig**), and it performs similarly to the chemostat model in parameter estimation accuracy (**Fig 4A and 4B**). The Wright–Fisher is less suitable when selection is strong ($s_C =$ 0.1), as the true parameters are not covered by the 50% or 95% HDR (**S8 Fig**). Nevertheless, estimation of the selection coefficient remains accurate, and the difference in estimation of the formation rate is less than an order of magnitude, with a 3- to 5-fold overestimation (MAP log ratio between 0.5 and 0.7) (**Fig 4C and 4D**).

## Inference using a set of observations

Our empirical dataset includes 11 biological replicates of the same evolution experiment. Differences in the dynamics between independent replicates may be explained by an underlying DFE rather than a single constant selection coefficient. It is possible to infer the DFE using all experiments simultaneously. However, inference of distributions from multiple experiments presents several challenges, common to other mixed-effects or hierarchical models [88]. Alternatively, individual values inferred from individual experiments could provide an approximation of the underlying DFE.

To test these 2 alternative strategies for inferring the DFE, we performed simulations in which we allowed for variation in the selection coefficient of *GAP1* CNVs for each population in a set of observations. We sampled 11 selection coefficients from a Gamma distribution with shape and scale parameters $\alpha$ and $\beta$, respectively, and an expected value $E(s) = \alpha\beta$ [69], and then simulated a single observation for each sampled selection coefficient. As the Wright–Fisher model is a suitable approximation of the chemostat model (**Fig 4**), we used the Wright–Fisher model both for generating our observation sets and for parameter inference.

For the observation sets, we used NPE to either infer a single selection coefficient for each observation or to directly infer the Gamma distribution parameters $\alpha$ and $\beta$ from all 11 observations. When inferring 11 selection coefficients, one for each observation in the observation set, we fit a Gamma distribution to 8 of the 11 inferred values (**Fig 5**, green lines). When directly inferring the DFE, we used a uniform prior for $\alpha$ from 0.5 to 15 and a log-uniform prior for $\beta$ from $10^{-3}$ to 0.8. We held out 3 experiments from the set of 11 and used a 3-layer neural network to reduce the remaining 8 observations to a 5-feature summary statistic vector, which we then used as an embedding net [71] with NPE to infer the joint posterior distribution of $\alpha$, $\beta$, and $\delta_C$ (**Fig 5**, blue lines). For each observation set, we performed each inference method 3 times, using different sets of 8 experiments to infer the underlying DFE.

We used Kullback–Leibler divergence to measure the difference between the true DFE and inferred DFE and find that the inferred selection coefficients from the single experiments capture the underlying DFE as well or better than direct inference of the DFE from a set of observations for both $\alpha = 1$ (an exponential distribution) and $\alpha = 10$ (sum of 10 exponentials)

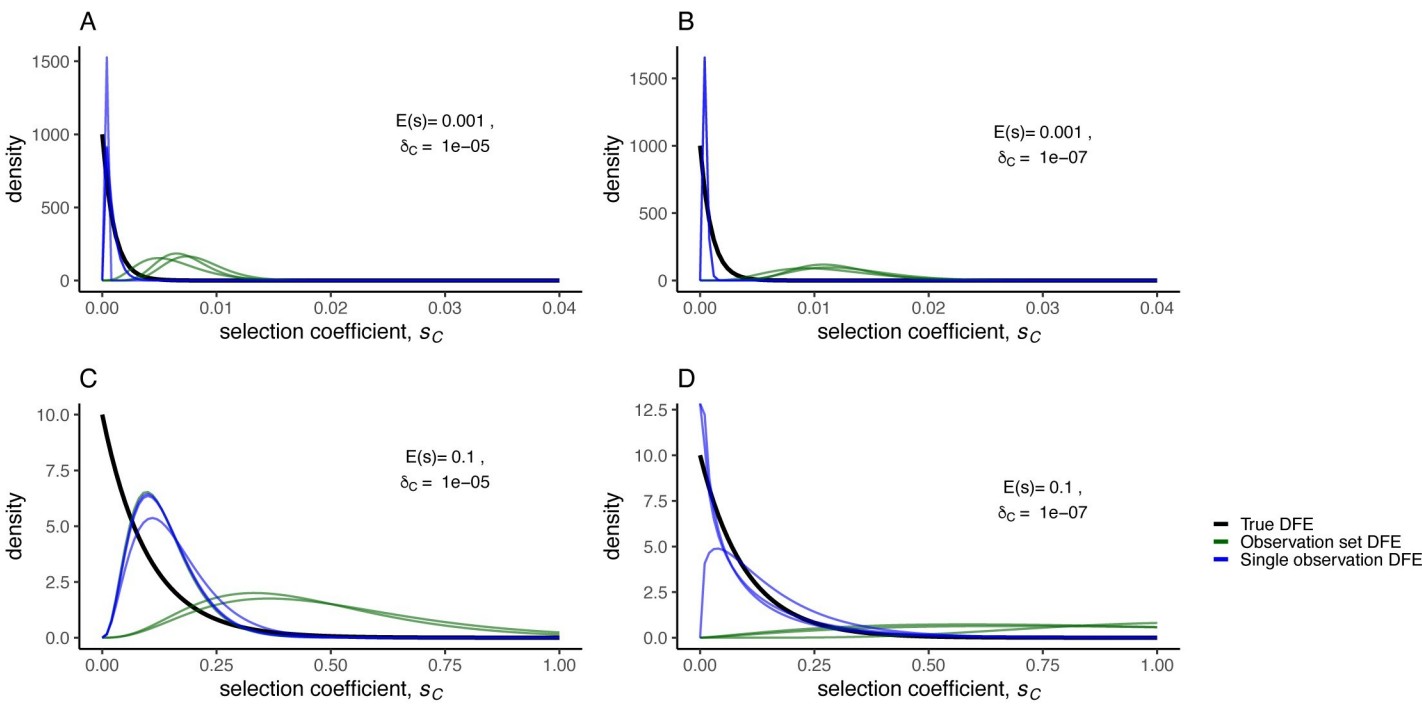

**Fig 5. Inference of the DFE.** A set of 11 simulated synthetic observations was generated from a WF model with CNV selection coefficients sampled from an exponential (Gamma with $\alpha = 1$) DFE (true DFE; black curve). The MAP DFEs (observation set DFE, green curves) were directly inferred using 3 different subsets of 8 out of 11 synthetic observations. We also inferred the selection coefficient for each individual observation in the set of 11 separately and fit a Gamma distribution (single observation DFE, blue curves) to sets of 8 inferred selection coefficients. All inferences were performed with NPE using the same amortized network to infer a posterior for each set of 8 synthetic observations or each single observation. **(A)** weak selection, high formation rate, **(B)** weak selection, low formation rate, **(C)** strong selection, high formation rate, **(D)** strong selection, low formation rate. Data and code required to generate this figure can be found at https://doi.org/10.17605/OSF.IO/E9D5X. CNV, copy number variant; DFE, distribution of fitness effects; MAP, maximum a posteriori; NPE, Neural Posterior Estimation; WF, Wright–Fisher.

(**Fig 5**, **S9 Fig**). The only exception we found is when $\alpha = 10$, $E(s) = 0.001$, and $\delta_C = 10^{-5}$ (**S9 Fig**, **S2 Table**). We assessed the performance of inference from a set of observations using out-of-sample posterior predictive accuracy [86] and found that inferring $\alpha$ and $\beta$ from a set of observations results in lower posterior predictive accuracy compared to inferring $s_C$ from a single observation (**S10 Fig**). Therefore, we conclude that estimating the DFE through inference of individual selection coefficients from each observation is superior to inference of the distribution from multiple observations.

## Inference from empirical evolutionary dynamics

To apply our approach to empirical data we inferred *GAP1* CNV selection coefficients and formation rates using 11 replicated evolutionary experiments in glutamine-limited chemostats [48] (**Fig 1A**) using NPE with both evolution models. We performed posterior predictive checks, drawing parameter values from the posterior distribution, and found that *GAP1* CNV were predicted to increase in frequency earlier and more gradually than is observed in our experimental populations (**S11 Fig**). This discrepancy is especially apparent in experimental populations that appear to experience clonal interference with other beneficial lineages (i.e., gln07, gln09). Therefore, we excluded data after generation 116, by which point CNVs have reached high frequency in the populations but do not yet exhibit the nonmonotonic and variable dynamics observed in later time points, and performed inference. The resulting posterior predictions are more similar to the observations in initial generations (average MAP RMSE for the 11 observations up to generation 116 is 0.06 when inference excludes late time points versus 0.13 when inference includes all time points). Furthermore, the overall RMSE (for observations up to generation 267) was not significantly different (average MAP RMSE is 0.129 and 0.126 when excluding or including late time points, respectively; **S12 Fig**). Restricting the analysis to early time points did not dramatically affect estimates of *GAP1* CNV selection coefficient and formation rate, but it did result in less variability in estimates between populations (i.e., independent observations) and some reordering of populations' selection coefficients and formation rate relative to each other (**S13 Fig**). Thus, we focused on inference using data prior to generation 116.

The inferred *GAP1* CNV selection coefficients were similar regardless of model, with the range of MAP estimates for all populations between 0.04 and 0.1, whereas the range of inferred *GAP1* CNV formation rates was somewhat higher when using the Wright–Fisher model, $10^{-4.1}$ to $10^{-3.4}$, compared to the chemostat model, $10^{-4.7}$ to $10^{-4}$ (**Fig 6A and 6B**). While there is variation in inferred parameters due to the training set, variation between observations (replicate evolution experiments) is higher than variation between training sets (**Fig 6A–6C**). Posterior predictions using the chemostat model, a fuller depiction of the evolution experiments, tend to have slightly lower RMSE than predictions using the Wright–Fisher model (**Fig 6C**). However, predictions using both models recapitulate actual *GAP1* CNV dynamics, especially in early generations (**Fig 6D**).

To test the sensitivity of these estimates, we also inferred the *GAP1* CNV selection coefficient and formation rate using the Wright–Fisher model in the absence of other beneficial mutations ($\delta_B = 0$), and for 9 additional combinations of other beneficial mutation selection coefficient $s_B$ and formation rate $\delta_B$ (**S14 Fig**). In general, perturbations to the rate and selection coefficient of other beneficial mutations did not alter the inferred *GAP1* CNV selection coefficient or formation rate. We found a single exception: When both the formation rate and fitness effect of other beneficial mutations is high ($s_B = 0.1$ and $\delta_B = 10^{-5}$), the *GAP1* CNV selection coefficient was approximately 1.6-fold higher and the formation rate was approximately 2-fold lower (**S14 Fig**); however, posterior predictions were poor for this set of parameter values (**S15 Fig**), suggesting that these values are inappropriate.

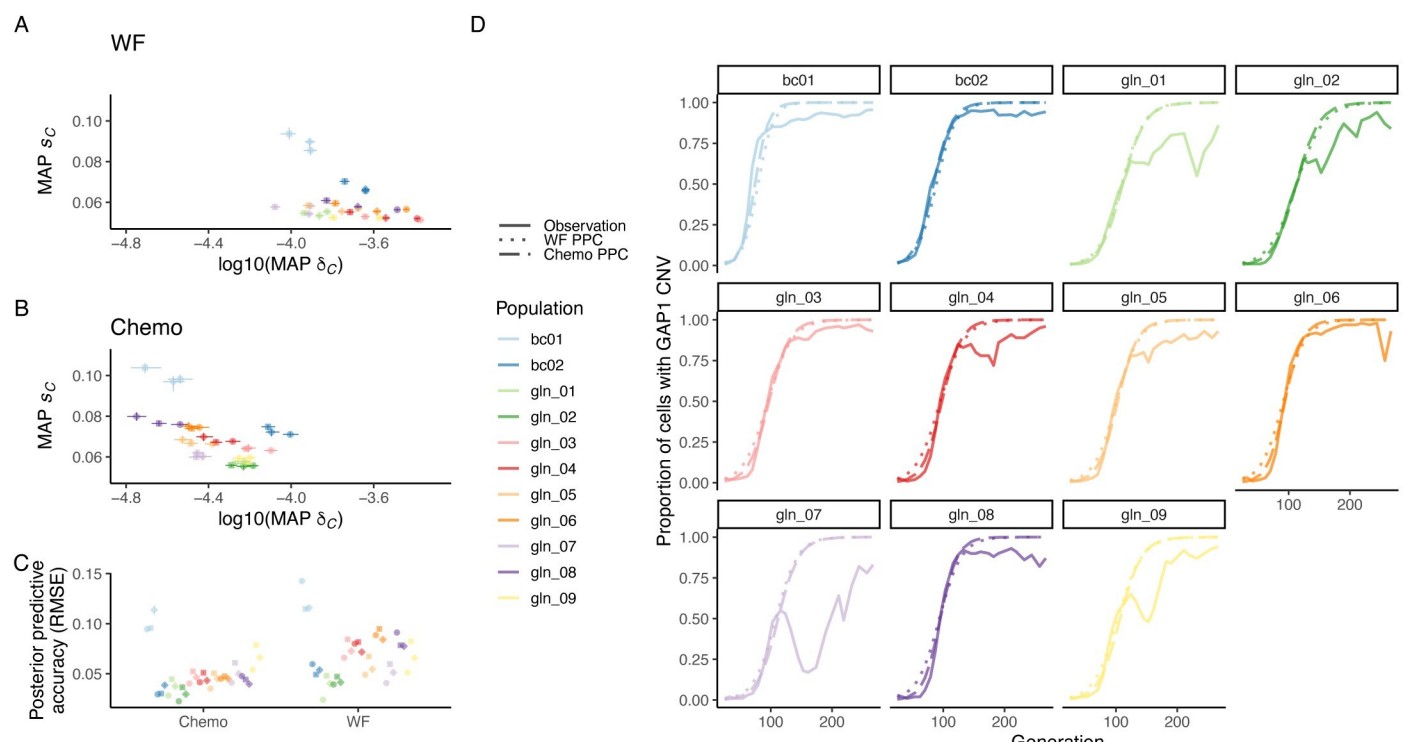

**Fig 6. Inference of CNV formation rate and fitness effect from empirical evolutionary dynamics.** The inferred MAP estimate and 95% HDIs for fitness effect $s_C$ and formation rate $\delta_C$, using the **(A)** WF or **(B)** chemostat (Chemo) model and NPE for each experimental population from [48]. Inference performed with data up to generation 116, and each training set (marker shape) corresponds to an independent amortized posterior distribution estimated with 100,000 simulations. **(C)** Mean and 95% confidence interval for RMSE of 50 posterior predictions compared to empirical observations up to generation 116. **(D)** Proportion of the population with a *GAP1* CNV in the experimental observations (solid lines) and in posterior predictions using the MAP estimate from one of the training sets shown in panels A and B with either the WF (dotted line) or chemostat (dashed line) model. Formation rate and fitness effect of other beneficial mutations set to $10^{-5}$ and $10^{-3}$, respectively. Data and code required to generate this figure can be found at https://doi.org/10.17605/OSF.IO/E9D5X. CNV, copy number variant; HDI, highest density interval; MAP, maximum a posteriori; NPE, Neural Posterior Estimation; RMSE, root mean square error; WF, Wright–Fisher.

## Experimental confirmation of fitness effects inferred from adaptive dynamics

To experimentally validate the inferred selection coefficients, we used lineage tracking to estimate the DFE [7,89,90]. We performed barseq on the entire evolving population at multiple time points and identified lineages that did and did not contain *GAP1* CNVs (**Fig 7A**). Using barcode trajectories to estimate fitness effects ([89]; see **Methods**), we identified 1,569 out of 80,751 lineages (1.94%) as adaptive in the bc01 population. A total of 1,513 (96.4%) adaptive lineages have a *GAP1* CNV (**Fig 7A**).

As a complementary experimental approach, selection coefficients can be directly measured using competition assays by fitting a linear model to the log ratio of the *GAP1* CNV strain and ancestral strain frequencies over time (**Fig 7B**). Therefore, we isolated *GAP1* CNV containing clones from populations bc01 and bc02, determined their fitness (**Methods**), and combined these estimates with previously reported selection coefficients for *GAP1* CNV containing clones isolated from populations gln01-gln09 [48] to define the DFE.

The DFE for adaptive *GAP1* CNV lineages in bc01 inferred using lineage-tracking barcodes and the DFE from pairwise competition assays share similar properties to the distribution inferred using NPE from all experimental populations (**Fig 7C**). Thus, our inference framework using CNV adaptation dynamics is a reliable estimate of the DFE estimated using laborious experimental methods that are gold standards in the field.

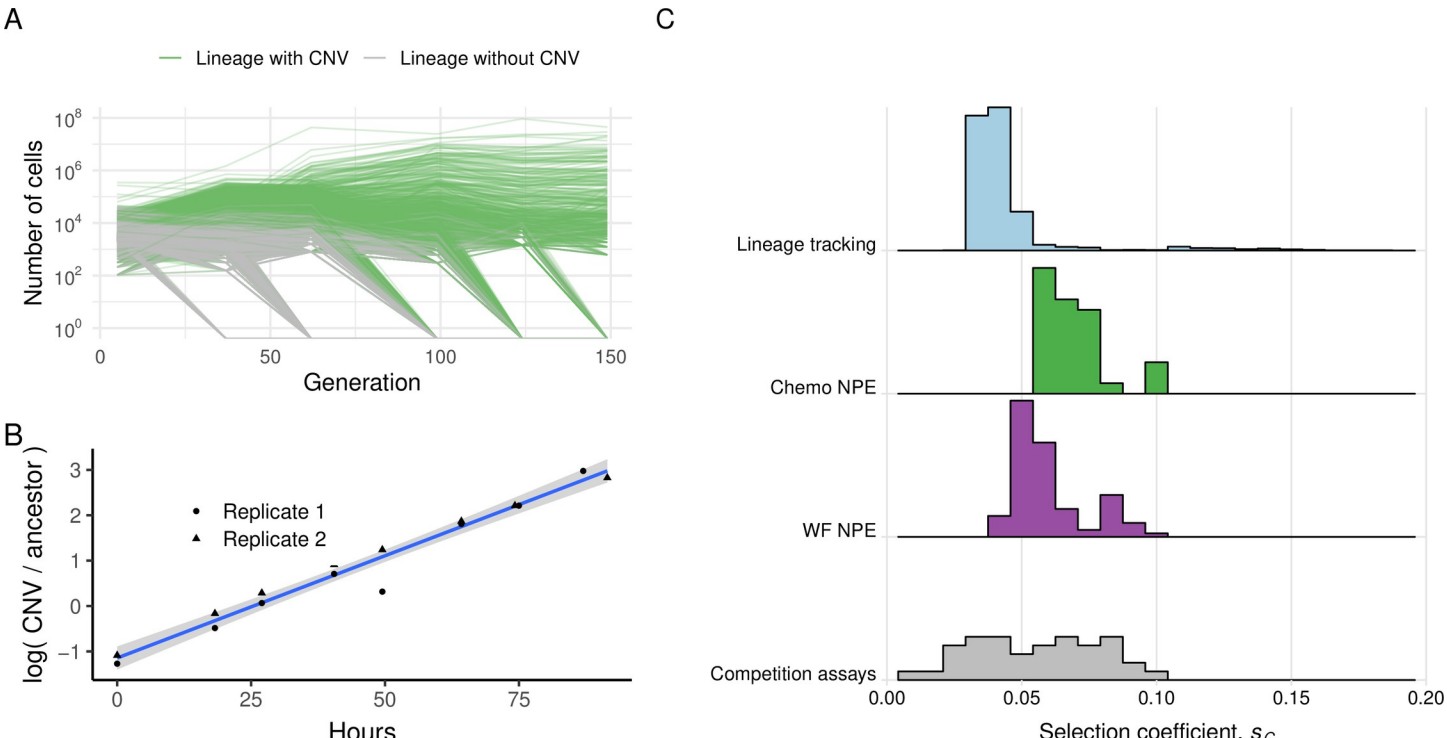

**Fig 7. Comparison of DFE inferred using NPE, lineage-tracking barcodes, and competition assays.** (A) Barcode-based lineage frequency trajectories in experimental population bc01. Lineages with (green) and without (gray) *GAP1* CNVs are shown. (B) Two replicates of a pairwise competition assay for a single *GAP1* CNV containing lineage isolated from an evolving population. The selection coefficient for the clone is estimated from the slope of the linear model (blue line) and 95% CI (gray). (C) The DFE for all beneficial *GAP1* CNVs inferred from 11 populations using NPE and the WF (purple) and chemostat (Chemo; green) models compared with the DFE inferred from barcode frequency trajectories in the bc01 population (light blue) and the DFE inferred using pairwise competition assays with different *GAP1* CNV containing clones (gray). Data and code required to generate this figure can be found at https://doi.org/10.17605/OSF.IO/E9D5X. CNV, copy number variant; DFE, distribution of fitness effects; NPE, Neural Posterior Estimation; WF, Wright–Fisher.

## Discussion

In this study, we tested the application of simulation-based inference for determining key evolutionary parameters from observed adaptive dynamics in evolution experiments. We focused on the role of CNVs in adaptive evolution using experimental data in which we quantified the population frequency of de novo CNVs at a single locus using a fluorescent CNV reporter. The goal of our study was to test a new computational framework for simulation-based, likelihood-free inference, compare it to the state-of-the-art method, and apply it to estimate the *GAP1* CNV selection coefficient and formation rates in experimental evolution using glutamine-limited chemostats.

Our study yielded several important methodological findings. Using synthetic data, we tested 2 different algorithms for joint inference of evolutionary parameters, the effect of different evolutionary models on inference performance, and how best to determine a DFE using multiple experiments. We find that the neural network–based algorithm NPE outperforms ABC-SMC regardless of evolutionary model. Although a more complex evolutionary model better describes the evolution experiments performed in chemostats, we find that a standard Wright–Fisher model can be a sufficient approximation for inference using NPE. However, the inferred *GAP1* CNV formation rate under the Wright–Fisher model is higher than under the chemostat model (**Fig 6A and 6B**), which is consistent with the overprediction of formation rates using the Wright–Fisher model for inference when an observation is generated by

the chemostat model and selection coefficients are high (**Fig 4C and 4D**). This suggests that Wright–Fisher is not the best suited model to use in all real-world cases, in particular if many beneficial CNVs turn out to have strong selection coefficients. Finally, although it is possible to perform joint inference on multiple independent experimental observations to infer a DFE, we find that inference performed on individual experiments and post facto estimation of the distribution more accurately captures the underlying DFE.

Previous studies that applied likelihood-free inference to results of evolutionary experiments differ from our study in various ways [5,6,49]. First, they used serial dilution rather than chemostat experiments. Second, most focused on all beneficial mutations, whereas we categorize beneficial mutations into 2 categories: *GAP1* CNVs and all other beneficial mutations; thus, they used an evolutionary model with a single process generating genetic variation, whereas our study includes 2 such processes, but focuses inference on our mutation type of interest. Third, we used 2 different evolutionary models: the Wright–Fisher model, a standard model in evolutionary genetics, and a chemostat model. The latter is more realistic but also more computationally demanding. Fourth and importantly, previous studies applied relatively simple rejection ABC methods [5,6,49,69]. We applied 2 modern approaches: ABC with sequential Monte Carlo sampling [63], which is a computationally efficient algorithm for Bayesian inference, using an adaptive distance function [81]; and NPE [78–80] with NSF [84]. NPE approximates an amortized posterior distribution from simulations. Thus, it is more efficient than ABC-SMC, as it can estimate a posterior distribution for new observations without requiring additional training. This feature is especially useful when a more computationally demanding model is better (e.g., the chemostat model when selection coefficients are high). Our study is the first, to our knowledge, to use neural density estimation to apply likelihood-free inference to experimental evolution data.

Our application of simulation-based inference yielded new insights into the role of CNVs in adaptive evolution. Using a chemostat model we estimated *GAP1* CNV formation rate and selection coefficient from empirical population-level adaptive evolution dynamics and found that *GAP1* CNVs form at a rate of $10^{-4.7}$ to $10^{-4.0}$ per generation (approximately 1 in 10,000 cell divisions) and have selection coefficients of 0.04 to 0.1 per generation. We experimentally validated our inferred fitness estimates using barcode lineage tracking and pairwise competition assays and showed that simulation-based inference is in good agreement with the 2 different experimental methods. The formation rate that we have determined for *GAP1* CNVs is remarkably high. Locus-specific CNV formation rates are extremely difficult to determine and fluctuation assays have yielded estimates ranging from $10^{-12}$ to $10^{-6}$ [91–95]. Mutation accumulation studies have yielded genome-wide CNV rates of about $10^{-5}$ [32,37,38], which is an order of magnitude lower than our locus-specific formation rate. We posit 2 possible explanations for this high rate: (1) CNVs at the *GAP1* locus may be deleterious in most conditions, including the putative nonselective conditions used for mutation-selection experiments, and therefore underestimated in mutation accumulation assays due to negative selection; and (2) under nitrogen-limiting selective conditions, in which *GAP1* expression levels are extremely high, a mechanism of induced CNV formation may operate that increases the rate at which they are generated, as has been shown at other loci in the yeast genome [96, 97]. Empirical validation of the inferred rate of *GAP1* CNV formation in nitrogen-limiting conditions requires experimental confirmation.

This simulation-based inference approach can be readily extended to other evolution experiments. In this study, we performed inference of parameters for a single type of mutation. This approach could be extended to infer the rates and effects of multiple types of mutations simultaneously. For example, instead of assuming a rate and selection coefficient for other beneficial mutations and performing ex post facto analyses looking at the sensitivity of inference of *GAP1* CNV parameters in other beneficial mutation regimes, one could simultaneously infer

parameters for both of these types of mutations. As shown using our barcode-sequencing data, many CNVs arise during adaptive evolution, and previous studies have shown that CNVs have different structures and mechanisms of formation [48,98]. Inferring a single effective selection coefficient and formation rate is a current limitation of our study that could be overcome by inferring rates and effects for different classes of CNVs (e.g., aneuploidy versus tandem duplication). Inspecting conditional correlations in posterior distributions involving multiple types of mutations has the potential to provide insights into how interactions between different classes of mutations shape evolutionary dynamics.

The approach could also be applied to CNV dynamics at other loci, in different genetic backgrounds, or in different media conditions. Ploidy and diverse molecular mechanisms likely impact CNV formation rates. For example, rates of aneuploidy, which result from nondisjunction errors, are higher in diploid yeast than haploid yeast, and chromosome gains are more frequent than chromosome losses [37]. There is considerable evidence for heterogeneity in the CNV rate between loci, as factors including local sequence features, transcriptional activity, genetic background, and the external environment may impact the mutation spectrum. For example, there is evidence that CNVs occur at a higher rate near certain genomic features, such as repetitive elements [42], tRNA genes [99], origins of replication [100], and replication fork barriers [101].

Furthermore, this approach could be used to infer formation rates and selection coefficients for other types of mutations in different asexually reproducing populations; the empirical data required is simply the proportion of the population with a given mutation type over time, which can efficiently be determined using a phenotypic marker, or similar quantitative data such as whole-genome whole-population sequencing. Evolutionary models could be extended to more complex evolutionary scenarios including changing population sizes, fluctuating selection, and changing ploidy and reproductive strategy, with an ultimate goal of inferring their impact on a variety of evolutionary parameters and predicting evolutionary dynamics in complex environments and populations. Applications to tumor evolution and viral evolution are related problems that are likely amenable to this approach.

## Methods

All source code and data for performing the analyses and reproducing the figures is available at https://doi.org/10.17605/OSF.IO/E9D5X. Code is also available at https://github.com/graceave/cnv_sims_inference.

### Evolutionary models

We modeled the adaptive evolution from an isogenic asexual population with frequencies $X_A$ of the ancestral (or wild type) genotype, $X_C$ of cells with a *GAP1* CNV, and $X_B$ of cells with a different type of beneficial mutation. Ancestral cells can gain a *GAP1* CNV or another beneficial mutation at rates $\delta_C$ and $\delta_B$, respectively. Therefore, the frequencies of cells of different genotypes after mutation are

$$x^{\dagger}_A = (1 - \delta_B - \delta_C)x_A,$$

$$x^{\dagger}_B = x_A \delta_B + x_B,$$

$$x^{\dagger}_C = x_A \delta_C + x_C$$

For simplicity, this model neglects cells with multiple mutations, which is reasonable for short timescales, such as those considered here.

In the discrete time Wright–Fisher model, the change in frequency due to natural selection is modeled by

$$x^*_i = \frac{w_k x_i}{\bar{w}}, \quad \bar{w} = \sum_{i \in \{A,B,C\}} w_i x_i,$$

where $w_i$ is the relative fitness of cells with genotype i, and $\bar{w}$ is the population mean fitness relative to the ancestral type. Relative fitness is related to the selection coefficient by

$$w_i = 1 + s_i, i = B, C$$

The change in frequency due random genetic drift is given by

$$n_i = Multinomial(N, (x^*_A, x^*_B, x^*_C)), \quad x'_i = \frac{n_i}{N},$$

where $N$ is the population size. In our simulations $N = 3.3 \times 10^8$, the effective population size in the chemostat populations in our experiment (see the "Determining the effective population size in the chemostat" section).

The chemostat model starts with a population size $1.5 \times 10^{-7}$ and the concentration of the limiting nutrient in the growth vessel, $S$, is equal to the concentration of that nutrient in the fresh media, $S_0$. During continuous culture, the chemostat is continuously diluted as fresh media flows in and culture media and cells are removed at rate $D$. During the initial phase of growth, the population size grows, and the limiting nutrient concentration is reduced until a steady state is attained at which the population size and limiting nutrient concentration are maintained indefinitely. We extended the model for competition between 2 haploid clonal populations for a single growth-limiting resource in a chemostat from [73] to 3 populations such that

$$\frac{dx_A}{dt} = x_A \left( \frac{r_A S}{S + k_A} - D \right),$$

$$\frac{dx_B}{dt} = x_B \left( \frac{r_B S}{S + k_B} - D \right),$$

$$\frac{dx_C}{dt} = x_C \left( \frac{r_C S}{S + k_C} - D \right),$$

$$\frac{dS}{dt} = (S_0 - S)D - \frac{x_A r_A S}{(S + k_A) Y_A} - \frac{x_B r_B S}{(S + k_B) Y_B} - \frac{x_C r_C S}{(S + k_C) Y_C}$$

$Y_i$ is the culture yield of strain i per mole of limiting nutrient. $r_A$ is the Malthusian parameter, or intrinsic rate of increase, for the ancestral strain, and in the chemostat literature is frequently referred to as $\mu_{max}$, the maximal growth rate. The growth rate in the chemostat, $\mu$, depends on the the concentration of the limiting nutrient with saturating kinetics $\mu = \frac{\mu_{max} S}{k_s + S}$. $k_i$ is the substrate concentration at half-maximal $\mu$. $r_C$ and $r_B$ are the Malthusian parameters for strains with a CNV and strains with another beneficial mutation, respectively, and are related to the ancestral Malthusian parameter and selection coefficient by [102]

$$s_i = \frac{r_i - r_A}{r_A} ln2, i = B, C.$$

The values for the parameters used in the chemostat model are in Table 1.

We simulated continuous time in the chemostat using the Gillespie algorithm with $\tau$-leaping. Briefly, we calculate the rates of ancestral growth, ancestral dilution, CNV growth, CNV dilution, other mutant growth, other mutant dilution, mutation from ancestral to CNV, and mutation from ancestral to other mutant. For the next time interval $\tau$, we calculated the number of times each event occurs during the interval using the Poisson distribution. The limiting substrate concentration is then adjusted accordingly. These steps repeat until the desired number of generations is reached.

For the chemostat model, we began counting generations after 48 hours, which is approximately the amount of time required for the chemostat to reach steady state, and when we began recording generations in [48].

## Determining the effective population size in the chemostat

In order to determine the effective population size in the chemostat, and thus the population size to use in with the Wright–Fisher model, we determined the conditional variance of the allele frequency in the next generation $p'$ given the frequency in the current generation $p$ in the chemostat. To do this, we simulated a chemostat population with 2 neutral alleles with frequencies $p$ and $q$ ($p + q = 1$), which begin at equal frequencies, $p = q$. We allowed the simulation to run for 1,000 generations, recording the frequency $p$ at every generation, excluding the first 100 generations to ensure the population is at steady state. We then computed the conditional variance $Var(p'|p)$ in each generation and estimated the effective population size as (where $t = 900$ is the total number of generations) [103]:

$$N_e = \frac{p(1 - p)}{\frac{1}{t}\sum^t var(p\prime|p)}.$$

The estimated effective population size in our chemostat conditions is $3.3 \times 10^8$, which is approximately two-thirds of the census population size $N$ when the chemostat is at steady state.

## Inference methods

For inference using single observations, we used the proportion of the population with a *GAP1* CNV at 25 time points as our summary statistics and defined a log-uniform prior for the formation rate ranging from $10^{-12}$ to $10^{-3}$ and a log-uniform prior for the selection coefficient from $10^{-4}$ to 0.4.

For inference using sets of observation, we used a uniform prior for $\alpha$ from 0.5 to 15, a log-uniform prior for $\beta$ from $10^{-3}$ to 0.8, and a log-uniform prior for the formation rate ranging from $10^{-12}$ to $10^{-3}$. For use with NPE, we used a 3-layer sequential neural network with linear transformations in each layer and rectified linear unit as the activation functions to encode the observation set into 5 summary statistics, which we then used as an embedding net with NPE.

We applied ABC-SMC implemented in the Python package *pyABC* [70]. For inference using single observations, we used an adaptively weighted Euclidean distance function with the root mean square deviation as the scale function. For inference using a set of observations, we used the squared Euclidean distance as our distance metric. We used 100 samples from the prior for initial calibration before the first round, and a maximum acceptance rate of either 10,000 or 100,000 for both single observations and observation sets (i.e.,10,000 single observations or 10,000 sets of 11 observations). For the acceptance rate of 10,000, we started inference with 100 samples, had a maximum of 1,000 accepted samples per round, and a maximum of 10 rounds. For the acceptance rate of 100,000, we started inference with 1,000 samples, had a maximum of 10,000 accepted samples per round, and a maximum of 10 rounds. The exact number of samples from the proposal distribution during each round of sampling were

adaptively determined based on the shape of the current posterior distribution [82]. For inference of the posterior for each observation, we performed multiple rounds of sampling until either we reached the acceptance threshold $\varepsilon < = 0.002$ or 10 rounds were performed.

We applied NPE implemented in the Python package *sbi* [71] using a MAF [83] or a NSF [84] as a conditional density estimator that learns an amortized posterior density for single observations. We used either 10,000 or 100,000 simulations to train the network. To test the dependence of our results on the set of simulations used to learn the posterior, we trained 3 independent amortized networks with different sets of simulations generated from the prior and compared our resulting posterior distributions for each observation.

### Assessment of performance of each method with each model

To test each method, we simulated 5 populations for each combination of the following CNV formation rates and fitness effects: $s_C = 0.001$ and $\delta_C = 10^{-5}$; $s_C = 0.1$ and $\delta_C = 10^{-5}$; $s_C = 0.001$ and $\delta_C = 10^{-7}$; $s_C = 0.1$ and $\delta_C = 10^{-7}$, for both the Wright–Fisher model and the chemostat model, resulting in 40 total simulated observations. We independently inferred the CNV fitness effect and formation rate for each simulated observation 3 times.

We calculated the MAP estimate by first estimating a Gaussian kernel density estimate (KDE) using *SciPy* (*scipy.stats.gaussian_kde*) [104] with at least 1,000 parameter combinations and their weights drawn from the posterior distribution. We then found the maximum of the KDE (using *scipy.optimize.minimize* with the Nelder–Mead solver). We calculated the 95% HDIs for the MAP estimate of each parameter using *pyABC* (*pyabc.visualization.credible.compute_credible_interval*) [70].

We performed posterior predictive checks by simulating CNV dynamics using the MAP estimate as well as 50 parameter values sampled from the posterior distribution. We calculated RMSE and correlation to measure agreement of the 50 posterior predictions with the observation and report the mean and 95% confidence intervals for these measures. For inference on sets of observations, we calculated the RMSE and correlation coefficient between the posterior predictions and each of the 3 held out observations, and report the mean and 95% confidence intervals for these measures over all 3 held out observations.

We calculated AIC using the standard formula

$$AIC = -2log(p(y|\hat{\theta})) + 2k,$$

where $\hat{\theta}$ is the MAP estimate, $k = 2$ is the number of inferred parameters, $y$ is the observed data, and $p$ is the inferred posterior distribution. We calculated *Watanabe-AIC* or WAIC according to both commonly used formulas:

$$WAIC1 = -2\sum_{i=1}^{n} log\left(\frac{1}{S}\sum_{s=1}^{S} p(y_i|\theta^s)\right) + 2\sum_{i=1}^{n}\left(log\left(\frac{1}{S}\sum_{s=1}^{S} p(y_i|\theta^s)\right) - \frac{1}{S}\sum_{s=1}^{S} p(y_i|\theta^s)\right)$$

$$WAIC2 = -2\sum_{i=1}^{n} log\left(\frac{1}{S}\sum_{s=1}^{S} p(y_i|\theta^s)\right) + 2\sum_{i=1}^{n} V_{s=1}^{S}(logp(y_i|\theta^s)),$$

where $S$ is the number of draws from the posterior distribution, $\theta^s$ is a sample from the posterior, and $V_{s=1}^{S}$ is the posterior sample variance.

### Pairwise competitions

We isolated CNV-containing clones from the populations on the basis of fluorescence and performed pairwise competitions between each clone and an unlabeled ancestral (FY4) strain. We

also performed competitions between the ancestral *GAP1* CNV reporter strain, with and without barcodes. To perform the competitions, we grew fluorescent *GAP1* CNV clones and ancestral clones in glutamine-limited chemostats until they reached steady state [48]. We then mixed the fluorescent strains with the unlabeled ancestor in a ratio of approximately 1:9 and performed competitions in the chemostats for 92 hours or about 16 generations, sampling approximately every 2 to 3 generations. For each time point, at least 100,000 cells were analyzed using an Accuri flow cytometer to determine the relative abundance of each genotype. Previously, we established that the ancestral *GAP1* CNV reporter has no detectable fitness effect compared to the unlabeled ancestral strain [48]. However, the *GAP1* CNV reporter with barcodes does appear to have a slight fitness cost associated with it; therefore, we took slightly different approaches to determine the selection coefficient relative to the ancestral state depending on whether or not a *GAP1* CNV containing clone was barcoded. If a clone was not barcoded, we determined relative fitness using linear regression of the log ratio of the frequency of the 2 genotypes against the number of elapsed hours. If a clone was barcoded, relative fitness was computed using linear regression of the log ratio of the frequencies of the barcoded *GAP1* CNV-containing clone and the unlabeled ancestor, and the log ratio of the frequencies of the unevolved barcoded *GAP1* CNV reporter ancestor to the unlabeled ancestor against the number of elapsed hours, adding an additional interaction term for the evolved versus ancestral state. We converted relative fitness from per hour to generation by dividing by the natural log of 2.

## Barcode sequencing

In our prior study, populations with lineage tracking barcodes and the *GAP1* CNV reporter were evolved in glutamine-limited chemostats [48], and whole population samples were periodically frozen in 15% glycerol. To extract DNA, we thawed pelleted cells using centrifugation and extracted genomic DNA using a modified Hoffman–Winston protocol, preceded by incubation with zymolyase at 37˚C to enhance cell lysis [105]. We measured DNA quantity using a fluorometer and used all DNA from each sample as input to a sequential PCR protocol to amplify DNA barcodes which were then purified using a Nucleospin PCR clean-up kit, as described previously[48,89].

We measured fragment size with an Agilent TapeStation 2200 and performed qPCR to determine the final library concentration. DNA libraries were sequenced using a paired-end $2 \times 150$ bp protocol on an Illumina NovaSeq 6000 using an XP workflow. Standard metrics were used to assess data quality (Q30 and %PF). We used the Bartender algorithm with UMI handling to account for PCR duplicates and to cluster sequences with merging decisions based solely on distance except in cases of low coverage (<500 reads/barcode), for which the default cluster merging threshold was used [69]. Clusters with a size less than 4 or with high entropy (>0.75 quality score) were discarded. We estimated the relative abundance of barcodes using the number of unique reads supporting a cluster compared to total library size. Raw sequencing data is available through the SRA, BioProject ID PRJNA767552.

## Detecting adaptive lineages in barcoded clonal populations

To detect spontaneous adaptive mutations in a barcoded clonal cell population that is evolved for over time, we used a Python-based pipeline (which can be found at https://github.com/FangfeiLi05/PyFitMut) based on a previously developed theoretical framework [89]. The pipeline identifies adaptive lineages and infers their fitness effects and establishment time. In a barcoded population, a lineage refers to cells that share the same DNA barcode. For each lineage in the barcoded population, beneficial mutations continually occur at a total beneficial

mutation rate Ub, with fitness effect s, which results in a certain spectrum of fitness effects of mutations μ(s). If a beneficial mutant survives random drift and becomes large enough to grow deterministically (exponentially), we say that the mutation carried by the mutant has established. Here, we use Wright fitness s, which is defined as average number of additional t offspring of a cell per generation, that is, n(t) = n(0)·(1 + s), with n(t) being the total number of cells at generation t (can be nonintegers). Briefly, for each lineage, assuming that the lineage is adaptive (i.e., a lineage with a beneficial mutation occurred and established), then estimates of the fitness effect and establishment time of each lineage are made by random initialization, and the expected trajectory of each lineage is estimated and compared to the measured trajectory. Fitness effect and establishment time estimates are iteratively adjusted to better fit the observed data until an optimum is reached. At the same time, the expected trajectory of the lineage is also estimated assuming that the lineage is neutral. Finally, Bayesian inference is used to determine whether the lineage is adaptive or neutral. An accurate estimation of the mean fitness is necessary to detect mutations and quantify their fitness effects, but the mean fitness is a quantity that cannot be measured directly from the evolution. Rather, it needs to be inferred through other variables. Previously, the mean fitness was estimated by monitoring the decline of neutral lineages [89]. However, this method fails when there is an insufficient number of neutral lineages as a result of low sequencing read depth. Here, we instead estimate the mean fitness using an iterative method. Specifically, we first initialize the mean fitness of the population as zero at each sequencing time point, then we estimate the fitness effect and establishment time for adaptive mutations, then we recalculate the mean fitness with the optimized fitness and establishment time estimates, repeating the process for several iterations until the mean fitness converges.

## Supporting information

**S1 Table. Wall time to run one simulation.** Running time for a single WF simulation or a single chemostat simulation for each of the following parameter combinations on a 2019 MacBook Pro operating Mac OS Catalina 10.15.7 with a 2.6 GHz 6-Core Intel Core i7 processor. Code required to generate this table can be found at https://doi.org/10.17605/OSF.IO/E9D5X. WF, Wright–Fisher.
(CSV)

**S2 Table. Kullback–Leibler divergence for Gamma distributions fit from single inferred selection coefficients versus the true underlying DFE, or for directly inferred Gamma distributions versus the true underlying DFE.** Code required to generate this table can be found at https://doi.org/10.17605/OSF.IO/E9D5X. DFE, distribution of fitness effects.
(CSV)

**S1 Fig. Interpolation for bc01 and bc02.** Populations gln01-gln09 and bc01-bc02 have different time points—the gln populations have 25 time points in total, whereas the bc populations have 32 time points in total. Of these, 12 of the time points are the same in both populations. To match the time points in the gln populations, we interpolated from the 2 nearest time points in the bc populations (using pandas.DataFrame.interpolate("values")). This way, we can use the same data (same time points) for inference for all 11 populations so that we can use the same amortized NPE posterior to infer parameters for both gln populations and bc populations. Original bc data are shown as black dots, the matched data, with interpolated time points, is shown as red crosses. Data and code required to generate this figure can be found at https://doi.org/10.17605/OSF.IO/E9D5X. NPE, Neural Posterior Estimation.
(PNG)

**S2 Fig. Performance assessment of NPE with MAF using single simulated synthetic observations.** These show the results of inference on 5 simulated synthetic observations generated using either the WF or chemostat (Chemo) model (and inference performed with the same model) per combination of fitness effect $s_C$ and formation rate $\delta_C$. Here, we show the results of performing one training set with NPE with MAF using 100,000 simulations for training and using the same amortized network to infer a posterior for each replicate synthetic observation. **(A)** Percentage of true parameters within the 50% HDR. **(B)** Distribution of widths of the fitness effect $s_C$ 95% HDI calculated as the difference between the 97.5 percentile and 2.5 percentile, for each inferred posterior distribution. **(C)** Distribution of the number of orders of magnitude encompassed by the formation rate $\delta_C$ 95% HDI, calculated as difference of the base 10 logarithms of the 97.5 percentile and 2.5 percentile, for each inferred posterior distribution. **(D)** Log ratio MAP estimate as compared to true parameters for $s_C$ and $\delta_C$. Note that each panel has a different y-axis. **(E)** Mean and 95% confidence interval for RMSE of 50 posterior predictions as compared to the synthetic observation for which inference was performed. **(F)** RMSE of posterior prediction generated with MAP parameters as compared to the synthetic observation for which inference was performed. **(G)** Mean and 95% confidence interval for correlation coefficient of 50 posterior predictions compared to the synthetic observation for which inference was performed. **(H)** Correlation coefficient of posterior prediction posterior prediction generated with MAP parameters compared to the synthetic observation for which inference was performed. Data and code required to generate this figure can be found at https://doi.org/10.17605/OSF.IO/E9D5X. HDI, highest density interval; HDR, highest density region; MAF, masked autoregressive flow; MAP, maximum a posteriori; NPE, Neural Posterior Estimation; RMSE, root mean square error; WF, Wright–Fisher.
(PNG)

**S3 Fig. NPE with the WF model performs as well or better than other combinations of model and method.** Results of inference on 5 simulated single synthetic observations generated using either the WF or chemostat (Chemo) model (and inference performed with the same model) per combination of fitness effect $s_C$ and formation rate $\delta_C$. Here, we show the results of performing training with NPE with NSF using 100,000 simulations for training and using the same amortized network to infer a posterior for each replicate synthetic observation, or ABC-SMC when the training budget was 10,000. **(A)** RMSE (lower is better) of posterior prediction generated with MAP parameters as compared to the synthetic observation on which inference was performed. **(B)** Correlation coefficient (higher is better) of posterior prediction generated with MAP parameters compared to the synthetic observation on which inference was performed. **(C)** Mean and 95% confidence interval for correlation coefficient (higher is better) of 50 posterior predictions (sampled from the posterior distribution) compared to the synthetic observation on which inference was performed. Data and code required to generate this figure can be found at https://doi.org/10.17605/OSF.IO/E9D5X. ABC-SMC, Approximate Bayesian Computation with Sequential Monte Carlo; MAP, maximum a posteriori; NPE, Neural Posterior Estimation; RMSE, root mean square error; WF, Wright–Fisher.
(PNG)

**S4 Fig. NPE and WF have the lowest information criteria.** WAIC and AIC (lower is better) of models fitted on single synthetic observations using either the WF or chemostat (Chemo) model and either ABC-SMC or NPE for different combinations of fitness effect $s_C$ and formation rate $\delta_C$ with simulation budgets of 10,000 or 100,000 simulations per inference procedure (facets). We were unable to complete ABC-SMC with the chemostat model (red) when the training budget was 100,000 within a reasonable time frame. Data and code required to generate this figure can be found at https://doi.org/10.17605/OSF.IO/E9D5X. ABC-SMC,

Approximate Bayesian Computation with Sequential Monte Carlo; AIC, Akaike information criterion; NPE, Neural Posterior Estimation; WAIC, widely applicable information criterion; WF, Wright–Fisher.
(PNG)

**S5 Fig. NPE performs similar to or better than ABC-SMC for 8 additional parameter combinations.** The figure shows the results of inference on 5 simulated synthetic observations using the WF model per combination of fitness effect $s_C$ and formation rate $\delta_C$. Simulations and inference were performed using the same model. For NPE, each training set corresponds to an independently amortized posterior distribution trained on a different set of 100,000 simulations, with which each synthetic observation was evaluated to produce a separate posterior distribution. For ABC-SMC, each training set corresponds to independent inference procedures on each observation with a maximum of 100,000 total simulations accepted for each inference procedure and a stopping criteria of 10 iterations or $\varepsilon < = 0.002$, whichever occurs first. **(A)** The percent of true parameters within the 50% or 95% HDR of the inferred posterior distribution. The bar height shows the average of 3 training sets. **(B, C)** Distribution of widths of 95% HDI of the posterior distribution of the fitness effect $s_C$ (B) and CNV formation rate $\delta_C$ (C), calculated as the difference between the 97.5 percentile and 2.5 percentile, for each separately inferred posterior distribution. **(D)** Log ratio (relative error) of MAP estimate to true parameter for $s_C$ and $\delta_C$. Note the different y-axis ranges. A perfectly accurate MAP estimate would have a log ratio of zero. **(E)** Mean and 95% confidence interval for RMSE of 50 posterior predictions as compared to the synthetic observation for which inference was performed. **(F)** RMSE of posterior prediction generated with MAP parameters as compared to the synthetic observation for which inference was performed. **(G)** Mean and 95% confidence interval for correlation coefficient of 50 posterior predictions compared to the synthetic observation for which inference was performed. **(H)** Correlation coefficient of posterior prediction posterior prediction generated with MAP parameters compared to the synthetic observation for which inference was performed. Data and code required to generate this figure can be found at https://doi.org/10.17605/OSF.IO/E9D5X. ABC-SMC, Approximate Bayesian Computation with Sequential Monte Carlo; HDI, highest density interval; HDR, highest density region; MAP, maximum a posteriori; NPE, Neural Posterior Estimation; RMSE, root mean square error; WF, Wright–Fisher.
(PNG)

**S6 Fig. Effect of simulation budget on relative error of MAP estimate and width of HDIs.** For NPE, amortized posteriors were estimated using either 10,000 or 100,000 simulations, with which each synthetic observation was evaluated to produce a separate posterior distribution. For ABC-SMC, a posterior was independently inferred for each observation with a maximum of 10,000 or 100,000 total simulations accepted and a stopping criteria of 10 iterations or $\varepsilon < = 0.002$, whichever occurs first. The gray lines in **(A, D)** indicates a relative error of zero (i.e., no difference between MAP parameters and true parameters). **(D, E, F)** We were unable to complete ABC-SMC with the chemostat model (red) when the training budget was 100,000 within a reasonable time frame. Data and code required to generate this figure can be found at https://doi.org/10.17605/OSF.IO/E9D5X. ABC-SMC, Approximate Bayesian Computation with Sequential Monte Carlo; MAP, maximum a posteriori; NPE, Neural Posterior Estimation.
(PNG)

**S7 Fig. The cumulative number of simulations needed to estimate posterior distributions for multiple observations.** The x-axis shows the number of replicate simulated synthetic

observations for a combination of parameters, and the y-axis shows the cumulative number of simulations needed to infer posteriors for an increasing number of observations *(see the "Overview of inference strategies" section for more details)*, for observations with different combinations of CNV selection coefficient $s_C$ and CNV formation rate $\delta_C$ **(A–D)**. Each facet represents a total simulation budget for NPE, or the maximum number of accepted simulations for ABC-SMC. Since NPE uses amortization, a single amortized network is trained with 10,000 or 100,000 simulations, and that network is then used to infer posteriors for each observation (note that a single amortized network was used to infer posteriors for all parameter combinations.) For ABC-SMC, each observation requires a separate inference procedure to be performed individually, and not all generated simulations are accepted for posterior estimation; therefore, the number of simulations used for a single observation may be more than the acceptance threshold, and the number of simulations needed increases with the number of observations for which a posterior is inferred. Data and code required to generate this figure can be found at https://doi.org/10.17605/OSF.IO/E9D5X. ABC-SMC, Approximate Bayesian Computation with Sequential Monte Carlo; CNV, copy number variant; NPE, Neural Posterior Estimation.
(PNG)

**S8 Fig. Results of inference on 5 simulated synthetic observations generated using either the WF or chemostat (Chemo) model per combination of fitness effect $s_C$ and formation rate $\delta_C$.** We performed inference on each synthetic observation using both models. For NPE, each training set corresponds to an independent amortized posterior trained with 100,000 simulations, with which each synthetic observation was evaluated. **(A)** Percentage of true parameters within the 50% HDR. The bar height shows the average of 3 training sets. **(B)** Percentage of true parameters within the 95% HDR. The bar height shows the average of 3 training sets. Data and code required to generate this figure can be found at https://doi.org/10.17605/OSF.IO/E9D5X. HDR, highest density region; NPE, Neural Posterior Estimation; WF, Wright–Fisher.
(PNG)

**S9 Fig. A set of 11 simulated synthetic observations was generated from a WF model with CNV selection coefficients sampled from an Gamma distribution where $\alpha$ = 10 of fitness effects (DFE) (black curve).** The MAP DFEs (blue curves) were directly inferred using 3 different subsets of 8 out of 11 synthetic observations. We also inferred the selection coefficient for each observation in the set of 11 individually, and fit Gamma distributions to sets of 8 inferred selection coefficients (green curves). All inferences were performed with NPE using the same amortized network to infer a posterior for each set of 8 synthetic observations or each single observation. Data and code required to generate this figure can be found at https://doi.org/10.17605/OSF.IO/E9D5X. DFE, distribution of fitness effects; MAP, maximum a posteriori; NPE, Neural Posterior Estimation; WF, Wright–Fisher.
(PNG)

**S10 Fig.** Out-of-sample posterior predictive accuracy using RMSE (**A**) or correlation (**B**) using 3 held out observations when $\alpha$ and $\beta$ are directly inferred from the other 8 observations, for $\alpha$ = 1 or $\alpha$ = 10 (facets). Data and code required to generate this figure can be found at https://doi.org/10.17605/OSF.IO/E9D5X. RMSE, root mean square error.
(PNG)

**S11 Fig.** Proportion of the population with a *GAP1* CNV in the experimental observations (black) and in posterior predictions using the MAP estimate shown in panels A and B with either the WF or chemostat (Chemo) model. Inference was performed with all data up to

generation 267 (WF ppc 267, Chemo ppc 267), or excluding data after generation 116 (WF ppc 116, Chemo ppc 116). Formation rate and fitness effect of other beneficial mutations set to $10^{-5}$ and $10^{-3}$, respectively. Data and code required to generate this figure can be found at https://doi.org/10.17605/OSF.IO/E9D5X. MAP, maximum a posteriori; WF, Wright–Fisher.
(PNG)

**S12 Fig. MAP predictions have lower error when inference is performed using only up to generation 116 and are most accurate for the first 116 generations.** MAP posterior prediction RMSE when inference was performed excluding data after generation 116 (left) or using all data up to generation 267 (right). RMSE was calculated using either the first 116 generations or using up to generation 267 (x-axis). Data and code required to generate this figure can be found at https://doi.org/10.17605/OSF.IO/E9D5X. MAP, maximum a posteriori; RMSE, root mean square error.
(PNG)

**S13 Fig.** The inferred MAP estimate and 95% HDIs for fitness effect $s_C$ and formation rate $\delta_C$, using the **(A)** WF or **(B)** chemostat (Chemo) model and NPE for each experimental population from Lauer and colleagues (2018). Inference was either performed with data up to generation 116 or with all data, up to generation 267 (facets). Each training set corresponds to 3 independent amortized posterior distributions estimated with 100,000 simulations. Data and code required to generate this figure can be found at https://doi.org/10.17605/OSF.IO/E9D5X. HDI, highest density interval; MAP, maximum a posteriori; NPE, Neural Posterior Estimation; WF, Wright–Fisher.
(PNG)

**S14 Fig. Sensitivity analysis.** *GAP1* CNV formation rate and selection coefficient inferred using NPE with the WF model does not change considerably when other beneficial mutations have different selection coefficients $s_B$ and formation rates $\delta_B$, except when both $s_B$ and $\delta_B$ are high (purple). Data and code required to generate this figure can be found at https://doi.org/10.17605/OSF.IO/E9D5X. CNV, copy number variant; NPE, Neural Posterior Estimation; WF, Wright–Fisher.
(PNG)

**S15 Fig.** Mean and 95% confidence interval for RMSE **(A)** and correlation **(B)** of 50 posterior predictions compared to empirical observations up to generation 116, using posterior distributions inferred when other beneficial mutations have different selection coefficients $s_B$ and formation rates $\delta_B$. Data and code required to generate this figure can be found at https://doi.org/10.17605/OSF.IO/E9D5X. RMSE, root mean square error.
(PNG)

## Acknowledgments

We thank Uri Obolski, Ilia Kohanovski, Mark Siegal, Molly Przeworski, and members of the Gresham and Ram labs for discussions and comments.

## Author Contributions

**Conceptualization:** Grace Avecilla, David Gresham, Yoav Ram.

**Data curation:** Grace Avecilla.

**Formal analysis:** Grace Avecilla, Yoav Ram.

**Funding acquisition:** Grace Avecilla, David Gresham, Yoav Ram.

**Investigation:** Grace Avecilla, Fangfei Li, Yoav Ram.

**Methodology:** Grace Avecilla, Julie N. Chuong, Yoav Ram.

**Project administration:** Grace Avecilla, David Gresham, Yoav Ram.

**Resources:** Grace Avecilla.

**Software:** Grace Avecilla, Yoav Ram.

**Supervision:** Gavin Sherlock, David Gresham, Yoav Ram.

**Validation:** Grace Avecilla.

**Visualization:** Grace Avecilla.

**Writing – original draft:** Grace Avecilla.

**Writing – review & editing:** Grace Avecilla, David Gresham, Yoav Ram.

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
