## [Editor Report · Decision Letter 0]

11 Oct 2021

Dear David, 

Thank you for submitting your manuscript entitled "Simulation-based inference of evolutionary parameters from adaptation dynamics using neural networks" for consideration as a Research Article by PLOS Biology.

Your manuscript has now been evaluated by the PLOS Biology editorial staff, as well as by an academic editor with relevant expertise, and I'm writing to let you know that we would like to send your submission out for external peer review.

IMPORTANT: We note that although you submitted this paper as a regular Research Article, we think that it would much more appropriate to review it as a Methods and Resources paper. Please could you change the article type to "Methods and Resources" when you upload your additional metadata (see next paragraph)?

Once your full submission is complete, your paper will undergo a series of checks in preparation for peer review. Once your manuscript has passed the checks it will be sent out for review. 

If your manuscript has been previously reviewed at another journal, PLOS Biology is willing to work with those reviews in order to avoid re-starting the process. Submission of the previous reviews is entirely optional and our ability to use them effectively will depend on the willingness of the previous journal to confirm the content of the reports and share the reviewer identities. Please note that we reserve the right to invite additional reviewers if we consider that additional/independent reviewers are needed, although we aim to avoid this as far as possible. In our experience, working with previous reviews does save time. 

If you would like to send your previous reviewer reports to us, please specify this in the cover letter, mentioning the name of the previous journal and the manuscript ID the study was given, and include a point-by-point response to reviewers that details how you have or plan to address the reviewers' concerns. Please contact me at the email that can be found below my signature if you have questions. 

Please re-submit your manuscript within two working days, i.e. by Oct 13 2021 11:59PM.

Kind regards,

Roli

Roland Roberts

Senior Editor

PLOS Biology

rroberts@plos.org

---

## [Decision Letter · Decision Letter 1]

3 Dec 2021

Dear David,

Thank you for submitting your manuscript entitled "Simulation-based inference of evolutionary parameters from adaptation dynamics using neural networks" for review as a Methods and Resources paper by PLOS Biology. As with all papers reviewed by the journal, yours was assessed and discussed by the PLOS Biology editors, an academic editor with relevant expertise and in this case by three independent reviewers. Based on the reviews, I regret that we will not be pursuing this manuscript for publication in the journal.

You’ll see that while reviewer #2 seems positive, both reviewers #1 and #3 are concerned that the comparison between the two methods is problematical. Among other concerns, reviewer #1 thinks that your approach is unnecessarily complicated and rev #3 thinks that its usefulness is very limited. Both reviewers #1 and #3 suggested PLOS Comp Bio as a possible alternative venue for this study.

The reviews are attached and we hope they may help you, should you decide to revise the manuscript for submission elsewhere. I'm sorry that we can't be more positive on this occasion. 

While we cannot consider your manuscript for publication in PLOS Biology, we suggest that you consider transferring your manuscript to PLOS Computational Biology (two of the reviewers made this recommendation). The PLOS journals are editorially independent, so we cannot guarantee it will be reviewed there, and you would need to address the reviewers' concerns before consideration.

If you would like to submit your work to PLOS Computational Biology, which is editorially independent, please click the following link:

<DeepLinkData><DeepLinkTypeID>27</DeepLinkTypeID><peopleID>261898</peopleID><userSecurityID>78872ea5-2994-4669-9617-0c941690a8ae</userSecurityID><documentID>47479</documentID><revision>1</revision><manuscriptNumber>PBIOLOGY-D-21-02590</manuscriptNumber><docSecurityID>b7d4fabc-2c99-401d-93dd-abb58fce55de</docSecurityID></DeepLinkData>

If you do not wish to submit your work to PLOS Computational Biology, please click this link to decline: 

<DeepLinkData><DeepLinkTypeID>28</DeepLinkTypeID><peopleID>261898</peopleID><userSecurityID>78872ea5-2994-4669-9617-0c941690a8ae</userSecurityID><documentID>47479</documentID><revision>1</revision><manuscriptNumber>PBIOLOGY-D-21-02590</manuscriptNumber><docSecurityID>b7d4fabc-2c99-401d-93dd-abb58fce55de</docSecurityID></DeepLinkData>

Please note, you can log into the submission sites with the same login that you used to submit to this journal. 

Should you choose to transfer your submission, you will receive a confirmation email within 24-48 hours after accepting the transfer. If you have any questions, please feel free to contact the journal at plosbiology@plos.org.

I hope you have found this review process constructive and that you will consider publishing your work in PLOS in future. Thank you for your support of PLOS and of Open Access publishing.

Sincerely,

Roli

Roland Roberts

Senior Editor

PLOS Biology

rroberts@plos.org

REVIEWERS' COMMENTS:

Reviewer #1:

The following review comments are based upon version R1 of the manuscript.

Avecilla et al compare the results of two methods for evolutionary inference, as applied to learning the rate of generation, and the selective advantage, of copy number variants in a population. A method of neural posterior estimation (NPE) outperforms a method of Approximate Bayesian Computation (ABC). I have preliminary comments about the question addressed in this study, followed by comments about the manuscript itself.

Preliminary comments

P1: As I understand it, evolutionary inference involves fitting a model to data so as to estimate evolutionary parameters. This requires a choice of model e.g. Wright-Fisher, a distance function, which describes the closeness of the output of the model to the observed data, and a means via which the parameter space of the model can be explored.

P2: As I understand it, the difference between the ABC and NPE methods that are applied here is the manner in which they explore parameter space. The models used and the distance functions are the same in each case.

P3: The example problem given of inferring mutation rates and selection coefficients is relatively simple. The output from the model, describing the proportion of cells with GAP1 CNV, is one-dimensional, if time-dependent, while the space of inferred parameters is only two-dimensional, with a selection coefficient s_C, and a mutation rate \\delta_C.

P4: Given point P3, I would expect a very simple model to produce a decent solution to this problem. For example, taking each replicate in turn, I expect that it would be possible to propose initial values of {s_C, \\delta_C}, then iteratively change these values in the manner of a downhill optimisation (i.e. accepting new parameters that give smaller distances between the model and the data) so as to infer optimal parameters for each replicate.

Comments:

1. While I appreciate the value of ABC methods in cases where a likelihood function is intractable I am not convinced that this is true for a Wright-Fisher model. Where allele frequency data is collected from a population using genome sequencing of representative samples from a population, the output would be expected to approximate a binomial sample under perfect sequencing and sampling, or an overdispersed binomial sample given errors. Where short reads are collected from a large part of the genome that is subject to copy number variation, the number of reads from that region would be expected to approximate a Poisson distribution under perfect sampling, or in reality more likely an overdispersed Poisson model. The variances of distributions can be estimated via repeat sampling or variance from deterministic behaviour; under these circumstances, and given the desire to estimate a posterior distribution that reflects the extent of uncertainty in the data, is there an advantage to a likelihood-free approach?

2. The results section says that interpolation was used to match time points between experiments. I am concerned about the effect of this on the biological inferences made. When making estimates from these data (as opposed to testing on real data) could the methods be applied simply to the data?

3. I am concerned about what precisely can be learnt from the comparison of the two methods. While it is claimed that the NPE method outperforms the ABC method, and results are shown to demonstrate this, I don't understand why the ABC method does worse. Given the simplicity of the problem, what is going wrong? Could it be that:

i) The ABC method is equally as good, but is computationally less efficient? For example, running the ABC method for more iterations gave a better result. Is ABC bad at exploring parameter space, or just slow to converge?

ii) Both methods are equally good, but the specific implementation of the ABC method you are using is not a good one, for example in terms of not having been coded very well or in an efficient manner?

While I accept the results of the comparison, I am left unconvinced as to whether NPE is particularly good, or whether ABC is just extraordinarily bad. Further, while I am aware of the potential for neural networks in machine learning applications I am also unconvinced about whether this particular application is a case of using an overly-complicated technology to solve a very simple problem. Is there a reason why naive parameter optimisation would not work for these data?

Minor comments:

1. Where details are given about numbers of iterations and so forth, it would be valuable to have a measure of the actual time taken to run the optimisation i.e. how many minutes and on what sort of machine.

2. The abstract mentions yeast. Is this S. cerevisiae?

Reviewer #2:

This manuscript presents a thorough analysis of the use of likelihood-free methods to infer the formation rate and fitness effects of CNVs from adaptive evolution experiments in which a fluorescent reporter is used to quantify CNV dynamics. The authors show based on simulations that CNV formation rates and fitness effects can be determined accurately using likelihood-free inference on both Wright-Fisher and chemostat models, except if the CNV has a very low formation rate and selection coefficient. Neural posterior estimation (NPE) was found to outperform approximate Bayesian computation with sequential Monte Carlo (ABC-SMC) as a likelihood-free inference algorithm. Furthermore, the authors validate their approach on experimental data, showing that NPE inference under Wright-Fisher or chemostat models yields similar selection coefficient estimates as obtained through barcode-based lineage tracking and pairwise competition assays.

Overall, the study convincingly shows that NPE-based likelihood-free inference is well-suited to obtain CNV formation rates and fitness effects from chemostat adaptive evolution experiments using fluorescent reporters. These results also open perspectives for efficient determination of mutation rates and selection coefficients in other experimental evolution setups. In brief, the methodology outlined here may be very valuable to the experimental evolution community.

The setup of the study is well thought-out and the manuscript is well-written. I have no major comments on the analyses performed or their presentation, but I do have a couple of minor points I feel should be addressed:

- While performing inference on a set of observations, the authors conclude that estimating the DFE through inference of individual selection coefficients from each observation is superior to inference of the distribution from multiple observations. Any suggestions on why this might be ?

- While inferring the GAP1 CNV selection coefficient and formation rate from empirical data, the authors find that the inferred GAP1 CNV formation rate under the Wright-Fisher model is higher than under the chemostat model. It might be worth pointing out that this is consistent with the overprediction of formation rates under Wright-Fisher in the simulation results when selection coefficients are high. This does suggest that Wright-Fisher is not the best-suited model to use in all real-world cases, in particular if many beneficial CNVs would turn out to have strong selection coefficients. Maybe this should be mentioned more explicitly.

- Why was only bc01 used for lineage tracking and not bc02 ?

- Figure 6 : add labels for training sets

- 'The difference between any combination of model and method was less than 2 for δC=10-5 and sC=0.001' : 10-5 needs to be 10-7.

- reference to 'Figure 2C, Supplementary XX' and several references to 'Supplementary Files' need to be amended

- labels of Supp Fig 3 partially fall off page.

- '...as well or better than direct inference of the DFE from a set of observations for both α = 1 (an exponential distribution) and α = 10 (sum of ten exponentials) (Figure 5)' : refer to Supp Fig 6 here as well.

- Supp Fig 13 : I don't understand what μ(s) is exactly, how it was inferred from the data, and why this plot is a histogram rather than a dot plot.

- The formulas for the frequencies of different genotypes after mutation in the Evolutionary models section of the Methods contains some errors. δN in the first equation should be δB, xC in the second equation should be xB and xN in the third equation should be xC.

- It is unclear where the formula for estimating the effective population size in the chemostat comes from. There seems to be a factor 2 missing in front of var(p'|p) and the averaging likely applies to the entire fraction rather than only the denominator. Additionally, the harmonic mean may be preferable for averaging over generations rather than the arithmetic mean. 

- 'If a clone was barcoded, relative fitness using linear regression of the natural logarithm of the ratio of the barcoded GAP1 CNV containing clone to the unlabeled ancestor, and the natural logarithm of the ratio of the unevolved barcoded GAP1 CNV reporter ancestor to the unlabeled ancestor against the number of elapsed hours, adding an additional interaction term for the evolved versus ancestral state.' : something is missing in this sentence.

- 'For each lineage in the barcoded population, beneficial mutations continually occur at a total beneficial mutation rate Ub, with fitness effect s, with a certain spectrum of fitness effects of mutations —(s).' : something is off in this sentence.

Reviewer #3:

[identifies himself as Thomas Bataillon]

[see also the attached marked up version of the manuscript PDF]

General comments

The submitted ms focuses on a specific question within the field of the genetics of adaptation in (completely asexual ) populations: estimating the mutation rate and selective effect of mutation at a specific locus while treating the rest of the genome "separately". This is warranted especially if one knows in advance that most of the action can be traced back to the locus of interest. The methods developed here are heavily inspired by a specific "yeast settings" that looks promising to track evolution in real time but where it is difficult to see how the system can give broad insights and how the methods developed and tested here can generalize to other settings.

Major issues.

A. The element that I find most limiting currently is that the inference methods that are presented here only are estimating two parameters: the rate of mutation at a specific locus (here the gap1 CNV) and effect of a beneficial mutation at this locus. All inference is predicated on the assumption that all other important parameters are "known" independently . This includes at least three critical evolutionary parameters that are notoriously difficult to estimate from data:

- the genome wide mutation rates at other loci in the genome

- the distribution of fitness effects of these "other" mutations

- the effective size realized during the whole experiment.

This makes for a very idiosyncratic study where it is hard generalize / gain insights for to a broader range of situations in experimental evolution.

B. The methodology used for comparing inference methods (ABC versus NPE) and to appreciate how the different estimators are behaving is very difficult to follow ( scattered between an overview , a method and numerous figures and supplementary figure legends), but after reading through carefully a few points are really baffling me :

In short :

1. it is hard to see how the two methods ( SMS-ABC and NPE) are compared on equal foots (in term of amount of computation made available to ensure that methods return sensible approximate posterior distribution) (see numerous comments to the authors in the annotated PDF) . It seems that ABC-SMC typically needs at least 10^5 simulations to approximate sensibly posteriors but it seems that in the main figure ( figure3) only a tenth are allowed (if I understood the figure legend) .

2. It seems that the comparisons made are based on at best a handful (usually 5 max in some instance apparently 15 ) of truly independent datasets simulated on very specific scenarios. (see my numerous comments on Figure 3 and several suppl figures comparing the methods)

I think a disproportionate emphasis is put on claiming superiority of NPE relative to BAC-SMC . I really have no share on either method . I am open to the fact that NPE is potentially superior to ABC-SMC.. but I currently remain unconvinced (because of points 1 and 2 above).

C. Scoping of the method / state of the art

Overall, that there is too much space devoted to comparing these methods and what is best and the manuscript could benefit by placing the inference proposed here in a wider context of numerous similar studies inferring the DFE of beneficial mutations using different experimental settings (eg EMPIRIC or other yeast based method or a plethora of work on E coli and what we have learned about beneficial mutations using these systems).

I also think that the ms would be stronger if there is more space devoted to exploring how much the inference is robust to the strong assumptions that all three other nuisance parameters are known without error in advance .

I hope that the comments I am enclosing along directly on the PDF are also useful

Best regards

Thomas Bataillon

---

## [Editor Report · Decision Letter 2]

21 Jan 2022

Dear David,

Thank you for submitting your manuscript "Simulation-based inference of evolutionary parameters from adaptation dynamics using neural networks" for consideration as a Methods and Resources paper at PLOS Biology. As mentioned in our previous communication, we have discussed the points that you raised in your Appeal with the Academic Editor, and they agree that you should be given a chance to address the concerns raised by the reviewers. Thank for your patience over the holiday period.

As mentioned previously, you’ll see that while reviewer #2 seems positive, both reviewers #1 and #3 are concerned that the comparison between the two methods is problematical. Among other concerns, reviewer #1 thinks that your approach is unnecessarily complicated and rev #3 thinks that its usefulness is very limited. We would be very interested to see your responses to these criticisms.

In light of the reviews (below), we will not be able to accept the current version of the manuscript, but we would welcome re-submission of a much-revised version that takes into account the reviewers' comments. We cannot make any decision about publication until we have seen the revised manuscript and your response to the reviewers' comments. Your revised manuscript is also likely to be sent for further evaluation by the reviewers.

We expect to receive your revised manuscript within 3 months. 

**IMPORTANT - SUBMITTING YOUR REVISION**

*Re-submission Checklist*

*Published Peer Review*

*PLOS Data Policy*

*Blot and Gel Data Policy*

Sincerely,

Roli

Roland Roberts

Senior Editor

PLOS Biology

rroberts@plos.org

REVIEWERS' COMMENTS:

Reviewer #1:

The following review comments are based upon version R1 of the manuscript.

Avecilla et al compare the results of two methods for evolutionary inference, as applied to learning the rate of generation, and the selective advantage, of copy number variants in a population. A method of neural posterior estimation (NPE) outperforms a method of Approximate Bayesian Computation (ABC). I have preliminary comments about the question addressed in this study, followed by comments about the manuscript itself.

Preliminary comments

P1: As I understand it, evolutionary inference involves fitting a model to data so as to estimate evolutionary parameters. This requires a choice of model e.g. Wright-Fisher, a distance function, which describes the closeness of the output of the model to the observed data, and a means via which the parameter space of the model can be explored.

P2: As I understand it, the difference between the ABC and NPE methods that are applied here is the manner in which they explore parameter space. The models used and the distance functions are the same in each case.

P3: The example problem given of inferring mutation rates and selection coefficients is relatively simple. The output from the model, describing the proportion of cells with GAP1 CNV, is one-dimensional, if time-dependent, while the space of inferred parameters is only two-dimensional, with a selection coefficient s_C, and a mutation rate \\delta_C.

P4: Given point P3, I would expect a very simple model to produce a decent solution to this problem. For example, taking each replicate in turn, I expect that it would be possible to propose initial values of {s_C, \\delta_C}, then iteratively change these values in the manner of a downhill optimisation (i.e. accepting new parameters that give smaller distances between the model and the data) so as to infer optimal parameters for each replicate.

Comments:

1. While I appreciate the value of ABC methods in cases where a likelihood function is intractable I am not convinced that this is true for a Wright-Fisher model. Where allele frequency data is collected from a population using genome sequencing of representative samples from a population, the output would be expected to approximate a binomial sample under perfect sequencing and sampling, or an overdispersed binomial sample given errors. Where short reads are collected from a large part of the genome that is subject to copy number variation, the number of reads from that region would be expected to approximate a Poisson distribution under perfect sampling, or in reality more likely an overdispersed Poisson model. The variances of distributions can be estimated via repeat sampling or variance from deterministic behaviour; under these circumstances, and given the desire to estimate a posterior distribution that reflects the extent of uncertainty in the data, is there an advantage to a likelihood-free approach?

2. The results section says that interpolation was used to match time points between experiments. I am concerned about the effect of this on the biological inferences made. When making estimates from these data (as opposed to testing on real data) could the methods be applied simply to the data?

3. I am concerned about what precisely can be learnt from the comparison of the two methods. While it is claimed that the NPE method outperforms the ABC method, and results are shown to demonstrate this, I don't understand why the ABC method does worse. Given the simplicity of the problem, what is going wrong? Could it be that:

i) The ABC method is equally as good, but is computationally less efficient? For example, running the ABC method for more iterations gave a better result. Is ABC bad at exploring parameter space, or just slow to converge?

ii) Both methods are equally good, but the specific implementation of the ABC method you are using is not a good one, for example in terms of not having been coded very well or in an efficient manner?

While I accept the results of the comparison, I am left unconvinced as to whether NPE is particularly good, or whether ABC is just extraordinarily bad. Further, while I am aware of the potential for neural networks in machine learning applications I am also unconvinced about whether this particular application is a case of using an overly-complicated technology to solve a very simple problem. Is there a reason why naive parameter optimisation would not work for these data?

Minor comments:

1. Where details are given about numbers of iterations and so forth, it would be valuable to have a measure of the actual time taken to run the optimisation i.e. how many minutes and on what sort of machine.

2. The abstract mentions yeast. Is this S. cerevisiae?

Reviewer #2:

This manuscript presents a thorough analysis of the use of likelihood-free methods to infer the formation rate and fitness effects of CNVs from adaptive evolution experiments in which a fluorescent reporter is used to quantify CNV dynamics. The authors show based on simulations that CNV formation rates and fitness effects can be determined accurately using likelihood-free inference on both Wright-Fisher and chemostat models, except if the CNV has a very low formation rate and selection coefficient. Neural posterior estimation (NPE) was found to outperform approximate Bayesian computation with sequential Monte Carlo (ABC-SMC) as a likelihood-free inference algorithm. Furthermore, the authors validate their approach on experimental data, showing that NPE inference under Wright-Fisher or chemostat models yields similar selection coefficient estimates as obtained through barcode-based lineage tracking and pairwise competition assays.

Overall, the study convincingly shows that NPE-based likelihood-free inference is well-suited to obtain CNV formation rates and fitness effects from chemostat adaptive evolution experiments using fluorescent reporters. These results also open perspectives for efficient determination of mutation rates and selection coefficients in other experimental evolution setups. In brief, the methodology outlined here may be very valuable to the experimental evolution community.

The setup of the study is well thought-out and the manuscript is well-written. I have no major comments on the analyses performed or their presentation, but I do have a couple of minor points I feel should be addressed:

- While performing inference on a set of observations, the authors conclude that estimating the DFE through inference of individual selection coefficients from each observation is superior to inference of the distribution from multiple observations. Any suggestions on why this might be ?

- While inferring the GAP1 CNV selection coefficient and formation rate from empirical data, the authors find that the inferred GAP1 CNV formation rate under the Wright-Fisher model is higher than under the chemostat model. It might be worth pointing out that this is consistent with the overprediction of formation rates under Wright-Fisher in the simulation results when selection coefficients are high. This does suggest that Wright-Fisher is not the best-suited model to use in all real-world cases, in particular if many beneficial CNVs would turn out to have strong selection coefficients. Maybe this should be mentioned more explicitly.

- Why was only bc01 used for lineage tracking and not bc02 ?

- Figure 6 : add labels for training sets

- 'The difference between any combination of model and method was less than 2 for δC=10-5 and sC=0.001' : 10-5 needs to be 10-7.

- reference to 'Figure 2C, Supplementary XX' and several references to 'Supplementary Files' need to be amended

- labels of Supp Fig 3 partially fall off page.

- '...as well or better than direct inference of the DFE from a set of observations for both α = 1 (an exponential distribution) and α = 10 (sum of ten exponentials) (Figure 5)' : refer to Supp Fig 6 here as well.

- Supp Fig 13 : I don't understand what μ(s) is exactly, how it was inferred from the data, and why this plot is a histogram rather than a dot plot.

- The formulas for the frequencies of different genotypes after mutation in the Evolutionary models section of the Methods contains some errors. δN in the first equation should be δB, xC in the second equation should be xB and xN in the third equation should be xC.

- It is unclear where the formula for estimating the effective population size in the chemostat comes from. There seems to be a factor 2 missing in front of var(p'|p) and the averaging likely applies to the entire fraction rather than only the denominator. Additionally, the harmonic mean may be preferable for averaging over generations rather than the arithmetic mean.

- 'If a clone was barcoded, relative fitness using linear regression of the natural logarithm of the ratio of the barcoded GAP1 CNV containing clone to the unlabeled ancestor, and the natural logarithm of the ratio of the unevolved barcoded GAP1 CNV reporter ancestor to the unlabeled ancestor against the number of elapsed hours, adding an additional interaction term for the evolved versus ancestral state.' : something is missing in this sentence.

- 'For each lineage in the barcoded population, beneficial mutations continually occur at a total beneficial mutation rate Ub, with fitness effect s, with a certain spectrum of fitness effects of mutations —(s).' : something is off in this sentence.

Reviewer #3:

[identifies himself as Thomas Bataillon]

[see also the attached marked up version of the manuscript PDF]

General comments

The submitted ms focuses on a specific question within the field of the genetics of adaptation in (completely asexual ) populations: estimating the mutation rate and selective effect of mutation at a specific locus while treating the rest of the genome "separately". This is warranted especially if one knows in advance that most of the action can be traced back to the locus of interest. The methods developed here are heavily inspired by a specific "yeast settings" that looks promising to track evolution in real time but where it is difficult to see how the system can give broad insights and how the methods developed and tested here can generalize to other settings.

Major issues.

A. The element that I find most limiting currently is that the inference methods that are presented here only are estimating two parameters: the rate of mutation at a specific locus (here the gap1 CNV) and effect of a beneficial mutation at this locus. All inference is predicated on the assumption that all other important parameters are "known" independently . This includes at least three critical evolutionary parameters that are notoriously difficult to estimate from data:

- the genome wide mutation rates at other loci in the genome

- the distribution of fitness effects of these "other" mutations

- the effective size realized during the whole experiment.

This makes for a very idiosyncratic study where it is hard generalize / gain insights for to a broader range of situations in experimental evolution.

B. The methodology used for comparing inference methods (ABC versus NPE) and to appreciate how the different estimators are behaving is very difficult to follow ( scattered between an overview , a method and numerous figures and supplementary figure legends), but after reading through carefully a few points are really baffling me :

In short :

1. it is hard to see how the two methods ( SMS-ABC and NPE) are compared on equal foots (in term of amount of computation made available to ensure that methods return sensible approximate posterior distribution) (see numerous comments to the authors in the annotated PDF) . It seems that ABC-SMC typically needs at least 10^5 simulations to approximate sensibly posteriors but it seems that in the main figure ( figure3) only a tenth are allowed (if I understood the figure legend) .

2. It seems that the comparisons made are based on at best a handful (usually 5 max in some instance apparently 15 ) of truly independent datasets simulated on very specific scenarios. (see my numerous comments on Figure 3 and several suppl figures comparing the methods)

I think a disproportionate emphasis is put on claiming superiority of NPE relative to BAC-SMC . I really have no share on either method . I am open to the fact that NPE is potentially superior to ABC-SMC.. but I currently remain unconvinced (because of points 1 and 2 above).

C. Scoping of the method / state of the art

Overall, that there is too much space devoted to comparing these methods and what is best and the manuscript could benefit by placing the inference proposed here in a wider context of numerous similar studies inferring the DFE of beneficial mutations using different experimental settings (eg EMPIRIC or other yeast based method or a plethora of work on E coli and what we have learned about beneficial mutations using these systems).

I also think that the ms would be stronger if there is more space devoted to exploring how much the inference is robust to the strong assumptions that all three other nuisance parameters are known without error in advance .

I hope that the comments I am enclosing along directly on the PDF are also useful

Best regards

Thomas Bataillon

---

## [Decision Letter · Decision Letter 3]

7 Apr 2022

Dear David,

Thank you for submitting your revised Methods and Resources paper entitled "Simulation-based inference of evolutionary parameters from adaptation dynamics using neural networks" for publication in PLOS Biology. I have now obtained advice from one of the original reviewers and have discussed their comments with the Academic Editor.

Unfortunately reviewer #3 was unable to re-review, but the Academic Editor has assessed your responses and revisions and is broadly satisfied. In addition, the Academic Editor also added a further comment (see foot of this email), which you should treat as optional.

Based on the reviews, we will probably accept this manuscript for publication, provided you satisfactorily address the remaining points raised by reviewer #1 and the Academic Editor. Please also make sure to address the following data and other policy-related requests.

IMPORTANT: Please attend to the following:

a) Please flip the title around to incorporate an active verb. We suggest: "Neural networks allow simulation-based inference of evolutionary parameters from adaptation dynamics."

b) Please attend to the requests from reviewer #1 and the Academic Editor (you should treat the latter as optional).

c) Please address my Data Policy requests below; specifically, we need you to supply the numerical values underlying Figs 1A, 3ABCDE, 4ABCD, 5ABCD, 6ABCD, 7ABC, S1, S2ABCDEFGHI, S3ABC, S4ABC, S5ABCDEFGH, S6ABCDEF, S7ABCD, S8AB, S9ABCD, S10AB, S11, S12, S13AB, S14, S15AB. My understanding is that all of the data and code required for the main Figs is in the Github deposition, but I’m not so sure about the Supplementary Figs; please clarify and/or rectify.

d) Please also cite the location of the data clearly in each relevant main and supplementary Fig legend, e.g. if, for example, the Figs can all be generated using the data and code in your Gthub deposition, you should write something like “Data and code required to generate this Figure can be found in https://github.com/graceave/cnv_sims_inference”.

We expect to receive your revised manuscript within two weeks. 

*Published Peer Review History*

*Press*

Sincerely,

Roli

Senior Editor,

rroberts@plos.org,

PLOS Biology

DATA POLICY:

Regardless of the method selected, please ensure that you provide the individual numerical values that underlie the summary data displayed in the following figure panels as they are essential for readers to assess your analysis and to reproduce it: Figs 1A, 3ABCDE, 4ABCD, 5ABCD, 6ABCD, 7ABC, S1, S2ABCDEFGHI, S3ABC, S4ABC, S5ABCDEFGH, S6ABCDEF, S7ABCD, S8AB, S9ABCD, S10AB, S11, S12, S13AB, S14, S15AB. NOTE: the numerical data provided should include all replicates AND the way in which the plotted mean and errors were derived (it should not present only the mean/average values).

DATA NOT SHOWN?

REVIEWER'S COMMENTS:

Reviewer #1:

I thank the authors for their clarifications. I am happy that NPE performs better for the specific task in hand than does an ABC approach.

To clarify my thoughts on the simplicity or otherwise of the problem, I agree with what I think de Sousa et al are saying, that the general problem of inferring the distribution of adaptive mutations available to natural selection is a difficult task. In addressing this question, their work, alongside that of Hegreness et al., and Barrick et al., both provides some insight while having limitations. For example, Barrick et al use a model which aims to identify only the first mutation to arise within each population; this throws away part of the data, and is almost certainly biased towards identifying mutations of stronger effect.

What I am less convinced is a difficult task is the specific question of how to fit a two-parameter Wright-Fisher model to data describing the evolution of a system under the influence of a single adaptive variant. If the function being optimised is not too rugged, a variety of approaches should give a decent answer to this problem. I remain unconvinced that likelihood approaches are as bad as the reviewers suggest. For example, if the log of the likelihood were calculated, I would not expect zero likelihoods would be a huge problem. Likelihood-based methods do not prevent the calculation of measure of confidence; a likelihood function plotted over parameter space would look similar to the posterior distributions of Figure 2. Further, identifying an approximate likelihood model might not be an especially intractable problem; the likelihood of a copy number variant having a specific frequency within the population given an observation might not be strictly analytical, but seems far from impossible to model if we accept that any modelling process involves a degree of approximation.

The matter of the simplicity of the problem is a somewhat tangential critique of the work described in this manuscript. The revised manuscript is an improvement on the previous one, and I am happy that for this specific question the NPE algorithm provides a useful insight into the specific benefit of CNV formation in the GAP1 locus of yeast. However I am unclear about whether the specific problem chosen is the best one with which to demonstrate the potential of NPE to solve more difficult problems within evolutionary inference. I would want to see applications of NPE to problems with larger numbers of parameters involved before accepting more general claims of its value for evolutionary inference.

Minor comments:

I note the WF-ABC method (Foll 2015) as another example of using ABC methods to process data from evolutionary experiments.

COMMENTS FROM THE ACADEMIC EDITOR:

[identifies himself as Arjan de Visser]

In addition to the comments of reviewer #1, I would like to point the authors to recent work involving myself, where we used Wright-Fisher simulations together with machine learning methods to infer mutation rates and fitness effects of 3 major mutation classes (SNPs, indels and structural variants), that best explained the observed mean and variance of each the numbers of mutations of each type in 96 evolved genotypes (Schenk, Zwart et al. 2022 Nat Ecol Evol, early online). Our approach is an alternative to using marker trajectories, which they (and de Sousa et al., Hegreness et al. and Barrick et al.) use, which the authors may consider mentioning in the introduction.

---

## [Editor Report · Decision Letter 4]

14 Apr 2022

Dear David,

On behalf of my colleagues and the Academic Editor, Arjan de Visser, I'm pleased to say that we can in principle accept your Methods and Resources "Neural networks allow simulation-based inference of evolutionary parameters from adaptation dynamics" for publication in PLOS Biology, provided you address any remaining formatting and reporting issues. These will be detailed in an email that will follow this letter and that you will usually receive within 2-3 business days, during which time no action is required from you. Please note that we will not be able to formally accept your manuscript and schedule it for publication until you have completed any requested changes.

Best wishes,

Roli

Roland G Roberts, PhD 

Senior Editor 

PLOS Biology

rroberts@plos.org